# 🦙🦙🦙 HealthGPT: A Medical Large Vision-Language Model for Unifying Comprehension and Generation via Heterogeneous Knowledge Adaptation

Tianwei Lin [1 2]   Wenqiao Zhang [1]   Sijing Li [1]   Yuqian Yuan [1 2]   Binhe Yu [3]   Haoyuan Li [2]   Wanggui He [2]
Hao Jiang [2]   Mengze Li [4]   Xiaohui Song [1]   Siliang Tang [1]   Jun Xiao [1]   Hui Lin [1]   Yueting Zhuang [1]
Beng Chin Ooi [5]

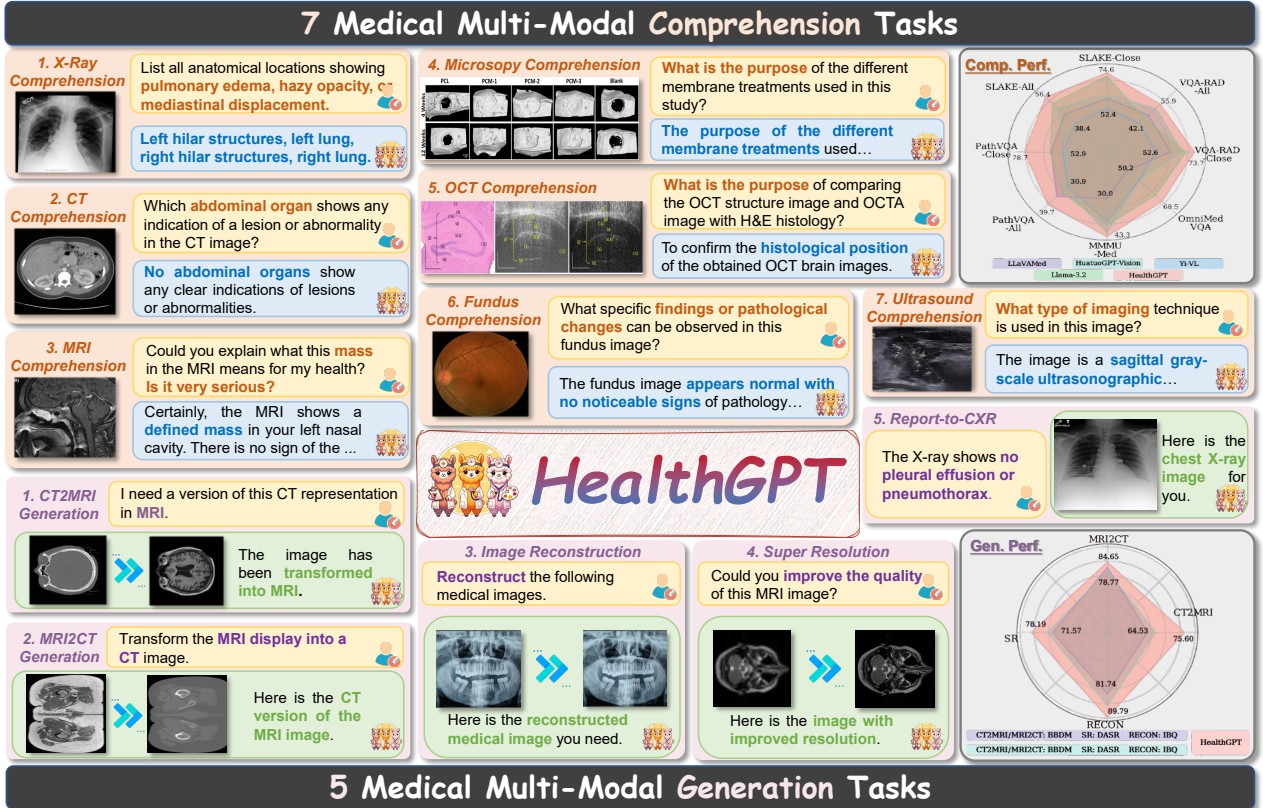

Figure 1: HealthGPT enables **medical multimodal comprehension and generation**, outperforming both state-of-the-art unified visual models and medical-specific models across various tasks. This highlights its superior capability in tackling complex tasks in healthcare applications. **Comp.Perf.** and **Gen.Perf.** denote the results of comprehension and generation.

## Abstract

We present **HealthGPT**, a powerful Medical Large Vision-Language Model (Med-LVLM) that integrates medical visual comprehension and generation capabilities within a unified autoregressive paradigm. Our bootstrapping philosophy is to progressively adapt heterogeneous comprehension and generation knowledge to pretrained Large Language Models (LLMs). This is achieved through a novel heterogeneous low-rank adaptation (**H-LoRA**) technique, which is complemented by a tailored hierarchical visual perception (**HVP**) approach and a three-stage learning strategy (**TLS**). To effectively learn the HealthGPT, we devise a comprehensive medical

[1]Zhejiang University, Hangzhou, China [2]Alibaba Group, Hangzhou, China [3]University of Electronic Science and Technology of China, Chengdu, China [4]The Hong Kong University of Science and Technology, HongKong, China [5]National University of Singapore, Singapore. Correspondence to: Wenqiao Zhang <wenqiaozhang@zju.edu.cn>.

*Proceedings of the 42nd International Conference on Machine Learning*, Vancouver, Canada. PMLR 267, 2025. Copyright 2025 by the author(s).

domain-specific comprehension and generation dataset called **VL-Health**. Experimental results demonstrate exceptional performance and scalability of HealthGPT in medical visual unified tasks. Our project can be accessed at https://github.com/DCDmllm/HealthGPT.

## 1. Introduction

Large Vision-Language Models (LVLMs) (Liu et al., 2023; OpenAI, 2023; Liu et al., 2024c; Chen et al., 2024b) have demonstrated outstanding open-world visual comprehension and reasoning abilities through language-based interactive dialogue over the past years, simultaneously opening up new opportunities for applications in specialized domains. Specifically, recent studies (Li et al., 2024b; Tu et al., 2024; Li et al., 2025) have utilized pre-trained Large Language Models (LLMs) and visual instruction data to build interactive diagnostic tools and treatment planning systems, revealing the immense potential of LVLMs in medical scenarios. However, these studies primarily concentrate on visual comprehension tasks that produce text-based outputs, such as medical visual question answering (Li et al., 2024b) or report generation (Nath et al., 2024), and deficient the "drawing" capability needed for medical visual generation. In practice, integrating visual comprehension and generation can significantly enhance the multifunctionality of medical LVLMs (Med-LVLMs).

Recent studies have increasingly focused on developing unified LVLMs capable of comprehending and generating content across diverse visual modalities. Earlier approaches predominantly utilized continuous visual tokens fed into LLMs, using the LLMs themselves as conditional generators for external generative models (Ge et al., 2024; Wu et al., 2024b; Dong et al., 2024). More recent research has explored the use of discrete visual tokens for image representation and generation within a fully autoregressive framework (Team, 2024a; Wang et al., 2024a; Xie et al., 2024). These methods not only enhance controllability but also demonstrate early success in open-world, any-to-any tasks, highlighting the preliminary potential of a unified autoregressive learning paradigm in multimodal tasks.

While unified LVLMs have achieved initial success in general scenarios, such a unified framework remains underexplored in the medical domain. Adapting the aforementioned general unified model paradigm to the medical domain presents two major challenges: **(i) High-scale and -quality Data Limitations**. Open-world models necessitate extensive pre-training on billions or even more diverse, multimodal data samples for comprehension and generation tasks (Lu et al., 2024; Team, 2024a). However, the accessible medical data significantly lacks in scale and quality compared to natural multimodal datasets. Its specialized and domain-specific characteristics make it challenging to

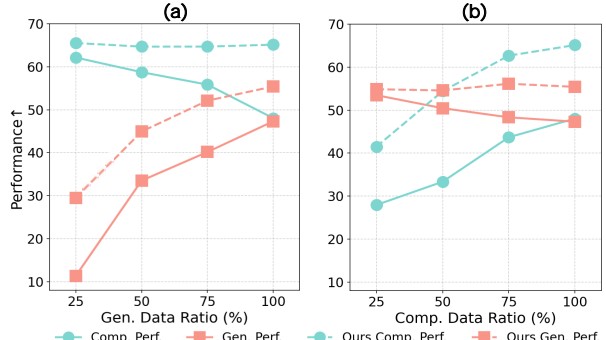

Figure 2: With a fixed amount of comprehension or generation data, increasing the proportion of the other type leads to significant performance degradation.

develop a unified medical model from scratch. **(ii) Conflicts between Comprehension and Generation**. Comprehension tasks often strip away visual details to focus on abstraction, while generation tasks require detailed preservation, making tokens sensitive to all visual alterations. As shown in Figure 2, which features experiments conducted on medical images, the performance in comprehension (or generation) tasks steadily decreases as the proportion of generation (or comprehension) data increases, and vice versa. This highlights a dilemma in autoregressive multimodal training, stemming from the need to maintain consistency of features between pre-LVLMs and post-LVLMs. Although some methods have explored mutual enhancement between comprehension and generation tasks (Pan et al., 2024; Tong et al., 2024), joint training in medical scenarios still faces challenges such as task conflict, data bias, and optimization saturation, leading to limited performance gains and persistent degradation issues.

To tackle the aforementioned challenges, we propose **HealthGPT** (see Figure 1), which progressively adapts a pre-trained LLM as an unified medical multimodal model with a small amount of visual instruction data. We devise innovative Parameter-Efficient Fine-Tuning (PEFT) approach (Ding et al., 2023), called **Heterogeneous Low-Rank Adaptation (H-LoRA)**, which decouples the learning process of LVLMs for comprehension and generation tasks. Inspired by the plug-and-play nature of LoRA (Hu et al., 2022), H-LoRA enables the model to store heterogeneous comprehension and generation knowledge in independent "plugins", thus avoiding joint optimization issues caused by conflicts between comprehension and generation tasks. In addition, we also consider the variety of sub-tasks among comprehension or generation tasks. Qualitative research highlights the limitations of a single LoRA in handling multidimensional task scenarios, mainly due to catastrophic forgetting and interference (Liu et al., 2024d; Lin et al., 2024). To address this, we draw on the concept of Mixture

of Experts (MoE) (Masoudnia & Ebrahimpour, 2014) and introduce LoRA experts. The aim is to dynamically transfer task-shared knowledge to adapt to downstream tasks. Unlike MoELoRA (Luo et al., 2024a), H-LoRA employs reversible matrix block multiplication to combine LoRA experts, significantly reducing the overhead of multiple matrix multiplications. **Notably, when using four experts, it requires only 67% of the MoELoRA training time.**

To effectively leverage H-LoRA in HealthGPT, we further introduce a **Hierarchical Visual Perception (HVP)** and devise a corresponding **Three-stage Learning Strategy (TLS)**. **HVP**: we separate visual details learning from Vision transformer (ViT) for comprehension and generation. As is widely recognized, the ViT encodes visual concepts with increasing abstraction, generally, becoming finer as we progress over levels (Vig, 2019). Thus, we maintain the visual features of the anterior and posterior layers to accommodate the differing requirements for visual granularity in comprehension and generation tasks while preventing potential task interference. **TLS**: In the first and second stages, given the heterogeneity between comprehension and generation tasks, we first train H-LoRA plugins for HealthGPT to incorporate both medical comprehension and generation knowledge, thus endowing the LLMs with capabilities for vision-language alignment and vision-to-vision reconstruction. Additionally, through minimal mixed-task training, we built fusion embedding layers and output heads that merge text and visual tokens, establishing a unified Med-LVLM foundation for visual instruction fine-tuning. In the third stage, by only training the H-LoRA plugins, HealthGPT is able to rapidly adapt to a wide range of downstream medical tasks, covering various types of medical comprehension and generation tasks.

To effectively implement our approach, we have curated a dataset for training unified Med-LVLMs, called **VL-Health**, including seven comprehension tasks and five generation tasks (Figure 1). Through quantitative analysis and validation on multimodal tasks, the results demonstrate that HealthGPT is capable of unifying medical multimodal abilities in data-constrained scenarios, achieving performance comparable to or better than existing state-of-the-art (SOTA) models across multiple metrics. Overall, the main contributions of this paper are summarized as follows:

- **Unified Med-LVLM.** We introduce HealthGPT, which, to the best of our knowledge, is the first unified framework for multimodal comprehension and generation in complex medical scenarios.

- **Effective Learning Paradigm.** We present H-LoRA, an optimized multi-LoRA PEFT architecture based on task-gated decoupling, is designed to effectively mitigate data conflict issues.

- **Holistic Training Dataset.** We curated VL-Health, a comprehensive dataset designed for both comprehension and generation tasks.

- **Superior Downstream Improvements**: Extensive experiments are conducted and the results confirm effectiveness of our approach in medical vision-language comprehension and generation.

## 2. Related Work

**Medical Vision Large Language Models.** In recent years, the development of deep learning brings prosperity to the field in medical applications (Zhang et al., 2022; Ong et al., 2022). Most recently, medical vision large language models (Med-VLLMs) demonstrate excellent performance in understanding medical images and responding to human queries based on these images (Zhou et al., 2023; Tian et al., 2023). XrayGPT (Thawakar et al., 2023) combines a medical visual encoder (MedClip) (Wang et al., 2022) with a fine-tuned LLM , using a simple linear transformation layer to achieve alignment between visual and textual information, significantly enhancing the understanding of medical images. On this basis, some methods (Wu et al., 2023; Li et al., 2024a; Bansal et al., 2025) further enhance visual-text alignment in medical contexts by selecting high-quality image-text pairs from PubMed papers and synthesized VQA datasets. BiomedGPT (Luo et al., 2024b) employs a BERT-style encoder and GPT-style decoder architecture, pre-trained on interdisciplinary datasets. Compared to commercial models like Med-PaLM (Singhal et al., 2023), BiomedGPT significantly reduces model size while maintaining superior performance. However, issues of language adaptability and dataset specificity still remain. To address these, HuatuoGPT-Vision (Chen et al., 2024a) introduces the PubMedVision dataset, which contains 1.3 million high-quality medical samples, significantly improving the model's adaptability across diverse medical applications. However, current Med-VLLMs mainly focus on medical comprehension and lack the capability for the medical vision-language generation.

**Unified Visual Comprehension and Generation Models.** Recent research has increasingly concentrated on creating unified LVLMs that are adept at understanding and producing content across various visual modalities. NExT-GPT (Wu et al., 2024b) achieves perception and generation for arbitrary combinations of multimodal inputs and outputs by aligning LLMs. Similarly, SEED (Ge et al., 2023), SEED-X (Ge et al., 2024), and DreamLLM (Dong et al., 2024) employ learnable queries and leverage next-token prediction to generate visual tokens, providing conditional inputs to external generation modules. Unlike these methods, which function as external conditioners, Unified-IO (Lu et al., 2022), Unified-IO 2 (Lu et al., 2024), and Chameleon (Team, 2024a) internalize multimodal generation tasks within a unified Transformer architecture by ex-

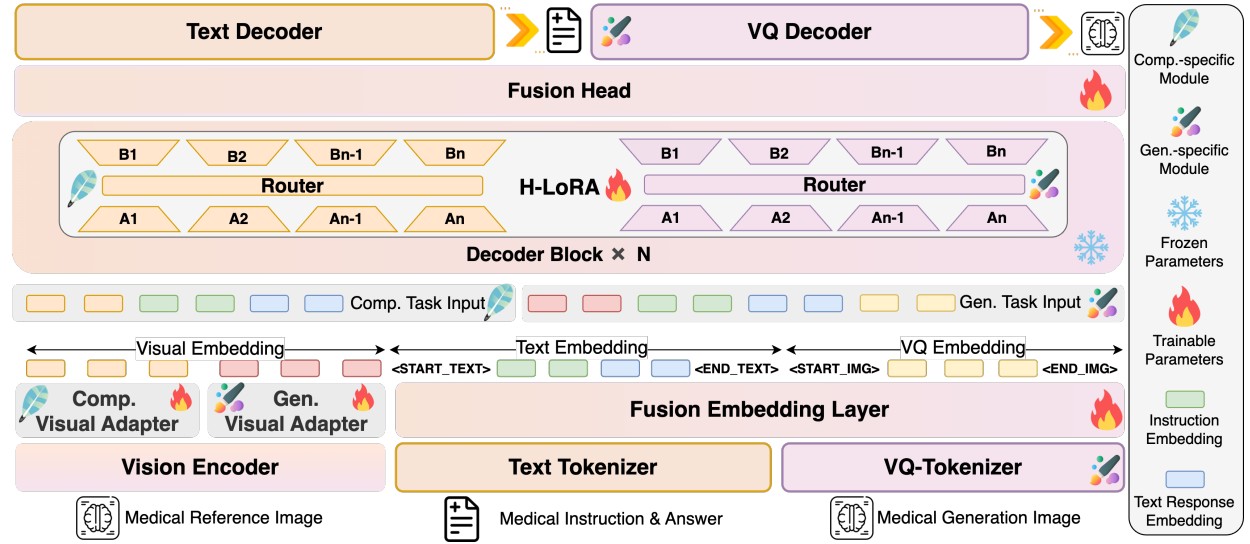

Figure 3: The HealthGPT architecture integrates hierarchical visual perception and H-LoRA, employing a task-specific hard router to select visual features and H-LoRA plugins, ultimately generating outputs with an autoregressive manner.

tending multimodal vocabularies, enabling direct generation based on next-token prediction. Building on this concept, Lumina-mGPT (Liu et al., 2024a) and ANOLE (Chern et al., 2024) further enhance the generation capabilities of unified models using high-quality data, particularly improving the quality and flexibility of image generation.

## 3. Preliminaries

**Large Vision-Language Models.** The input to a LVLM typically consists of an image $x^{\text{img}}$ and a discrete text sequence $x^{\text{txt}}$. The visual encoder $\mathcal{E}^{\text{img}}$ converts the input image $x^{\text{img}}$ into a sequence of visual tokens $\mathcal{V} = [v_i]_{i=1}^{N_v}$, while the text sequence $x^{\text{txt}}$ is mapped into a sequence of text tokens $\mathcal{T} = [t_i]_{i=1}^{N_t}$ using an embedding function $\mathcal{E}^{\text{txt}}$. The LVLM $\mathcal{M}_{\text{LVLM}}(\cdot|\theta)$ models the joint probability of the token sequence $\mathcal{U} = \{\mathcal{V}, \mathcal{T}\}$, which is expressed as:

$$P_\theta(R|\mathcal{U}) = \prod_{i=1}^{N_r} P_\theta(r_i|\{\mathcal{U}, r_{<i}\}), \quad (1)$$

where $R = [r_i]_{i=1}^{N_r}$ is the text response sequence. The LVLM iteratively generates the next token $r_i$ based on $r_{<i}$. The optimization objective is to minimize the cross-entropy loss of the response $\mathcal{R}$. It is worth noting that most LVLMs adopt a design paradigm based on ViT, alignment adapters, and pre-trained LLMs(Liu et al., 2023; 2024b), enabling quick adaptation to downstream tasks.

**VQGAN.** VQGAN (Esser et al., 2021) employs latent space compression and indexing mechanisms to effectively learn a complete discrete representation of images. VQGAN first maps the input image $x^{\text{img}}$ to a latent representation $z = \mathcal{E}(x)$ through a encoder $\mathcal{E}$. Then, the latent representation

is quantized using a codebook $\mathcal{Z} = \{z_k\}_{k=1}^{K}$, generating a discrete index sequence $\mathcal{I} = [i_m]_{m=1}^{N}$, where $i_m \in \mathcal{Z}$ represents the quantized code index:

$$\mathcal{I} = \text{Quantize}(z|\mathcal{Z}) = \arg\min_{z_k \in \mathcal{Z}} \|z - z_k\|_2. \quad (2)$$

In our approach, the discrete index sequence $\mathcal{I}$ serves as a supervisory signal for the generation task, enabling the model to predict the index sequence $\hat{\mathcal{I}}$ from input conditions such as text or other modality signals. Finally, the predicted index sequence $\hat{\mathcal{I}}$ is upsampled by the VQGAN decoder $G$, generating the high-quality image $\hat{x}^{\text{img}} = G(\hat{\mathcal{I}})$.

**Low Rank Adaptation.** LoRA(Hu et al., 2022) effectively captures the characteristics of downstream tasks by introducing low-rank adapters. The core idea is to decompose the bypass weight matrix $\Delta W \in \mathbb{R}^{d^{\text{in}} \times d^{\text{out}}}$ into two low-rank matrices $\{A \in \mathbb{R}^{d^{\text{in}} \times r}, B \in \mathbb{R}^{r \times d^{\text{out}}}\}$, where $r \ll \min\{d^{\text{in}}, d^{\text{out}}\}$, significantly reducing learnable parameters. The output with the LoRA adapter for the input $x$ is then given by:

$$h = xW_0 + \alpha x \Delta W/r = xW_0 + \alpha xAB/r, \quad (3)$$

where matrix $A$ is initialized with a Gaussian distribution, while the matrix $B$ is initialized as a zero matrix. The scaling factor $\alpha/r$ controls the impact of $\Delta W$ on the model.

## 4. HealthGPT

### 4.1. Unified Autoregressive Generation.

Our approach (Figure 3) utilizes a discrete token representation that covers both text and visual outputs, unifying visual

comprehension and generation as an autoregressive task. For comprehension, pre-trained LLM $\mathcal{M}_{\text{LLM}}(\cdot|\theta)$ receives the input joint sequence $\mathcal{U}$ and outputs a series of text token $\mathcal{R} = [\boldsymbol{r}_1, \boldsymbol{r}_2, \ldots, \boldsymbol{r}_{N_r}]$, where $\boldsymbol{r}_i \in \mathcal{V}_{\text{txt}}$, and $\mathcal{V}_{\text{txt}}$ represents the vocabulary of LLM:

$$P_\theta(\mathcal{R} \mid \mathcal{U}) = \prod_{i=1}^{N_r} P_\theta(\boldsymbol{r}_i \mid \mathcal{U}, \boldsymbol{r}_{<i}). \quad (4)$$

For generation, $\mathcal{M}_{\text{LLM}}$ first receives a special start token $\langle|\text{START\_IMG}|\rangle$, then generates a series of tokens corresponding to the VQGAN indices $\mathcal{I} = [\boldsymbol{i}_1, \boldsymbol{i}_2, \ldots, \boldsymbol{i}_{N_i}]$, where $\boldsymbol{i}_j \in \mathcal{V}_{\text{vq}}$, and $\mathcal{V}_{\text{vq}}$ represents the index range of VQGAN. Upon completion of generation, the LLM outputs an end token $\langle|\text{END\_IMG}|\rangle$:

$$P_\theta(\mathcal{I} \mid \mathcal{U}) = \prod_{j=1}^{N_i} P_\theta(\boldsymbol{i}_j \mid \mathcal{U}, \boldsymbol{i}_{<j}). \quad (5)$$

Finally, the generated index sequence $\mathcal{I}$ is fed into the decoder $G$, which reconstructs the target image $\hat{\boldsymbol{x}}^{\text{img}} = G(\mathcal{I})$.

### 4.2. Hierarchical Visual Perception

Given the differences in visual perception between comprehension and generation tasks — where the former focuses on abstract semantics and the latter emphasizes complete semantics—we employ ViT to compress the image into discrete visual tokens at multiple hierarchical levels. Specifically, the image is converted into a series of features $\{\boldsymbol{f}_1, \boldsymbol{f}_2, \ldots, \boldsymbol{f}_L\}$ as it passes through $L$ ViT blocks.

To address the needs of various tasks, the hidden states are divided into two types: (i) *Concrete-grained features* $\mathcal{F}^{\text{Con}} = \{\boldsymbol{f}_1, \boldsymbol{f}_2, \ldots, \boldsymbol{f}_k\}$ for $k < L$, derived from the shallower layers of ViT, containing sufficient global features, suitable for generation tasks; (ii) *Abstract-grained features* $\mathcal{F}^{\text{Abs}} = \{\boldsymbol{f}_{k+1}, \boldsymbol{f}_{k+2}, \ldots, \boldsymbol{f}_L\}$, derived from the deeper layers of ViT, which contain abstract semantic information closer to the text space, suitable for comprehension tasks.

The task type $T$ (comprehension or generation) determines which set of features is selected as the input for the downstream large language model:

$$\mathcal{F}_T^{\text{img}} = \begin{cases} \mathcal{F}^{\text{Con}}, & \text{if } T = \text{Generation Task} \\ \mathcal{F}^{\text{Abs}}, & \text{if } T = \text{Comprehension Task} \end{cases} \quad (6)$$

We integrate the image features $\mathcal{F}_T^{\text{img}}$ and text features $\mathcal{T}$ into a joint sequence through simple concatenation, which is then fed into the LLM $\mathcal{M}_{\text{LLM}}$ for autoregressive generation.

### 4.3. Heterogeneous Knowledge Adaptation

We devise H-LoRA, which stores heterogeneous knowledge from comprehension and generation tasks in separate

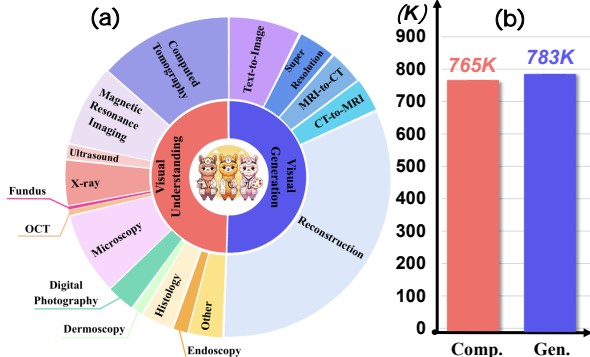

Figure 4: Data statistics of VL-Health.

modules and dynamically routes to extract task-relevant knowledge from these modules. At the task level, for each task type $T$, we dynamically assign a dedicated H-LoRA submodule $\theta^T$, which is expressed as:

$$\mathcal{R} = \mathcal{M}_{\text{LLM}}(\mathcal{U}|\theta, \theta^T), \quad \theta^T = \{\boldsymbol{A}^T, \boldsymbol{B}^T, \mathcal{R}_{\text{outer}}^T\}. \quad (7)$$

At the feature level for a single task, H-LoRA integrates the idea of Mixture of Experts (MoE) (Masoudnia & Ebrahimpour, 2014) and designs an efficient matrix merging and routing weight allocation mechanism, thus avoiding the significant computational delay introduced by matrix splitting in existing MoELoRA (Luo et al., 2024a). Specifically, we first merge the low-rank matrices (rank = $r$) of $k$ LoRA experts into a unified matrix:

$$\boldsymbol{A}^{\text{merged}}, \boldsymbol{B}^{\text{merged}} = \text{Concat}(\{\boldsymbol{A}_i\}_1^k), \text{Concat}(\{\boldsymbol{B}_i\}_1^k), \quad (8)$$

where $\boldsymbol{A}^{\text{merged}} \in \mathbb{R}^{d^{\text{in}} \times rk}$ and $\boldsymbol{B}^{\text{merged}} \in \mathbb{R}^{rk \times d^{\text{out}}}$. The $k$-dimension routing layer generates expert weights $\mathcal{W} \in \mathbb{R}^{\text{token\_num} \times k}$ based on the input hidden state $\boldsymbol{x}$, and these are expanded to $\mathbb{R}^{\text{token\_num} \times rk}$ space as follows:

$$\mathcal{W}^{\text{expanded}} = \alpha k \mathcal{W}/r \otimes \mathbf{1}_r, \quad (9)$$

where $\otimes$ denotes the replication operation. The overall output of H-LoRA is computed as:

$$\mathcal{O}^{\text{H-LoRA}} = (\boldsymbol{x}\boldsymbol{A}^{\text{merged}} \odot \mathcal{W}^{\text{expanded}})\boldsymbol{B}^{\text{merged}}, \quad (10)$$

where $\odot$ represents element-wise multiplication. Finally, the output of H-LoRA is added to the output of frozen pre-trained weights to produce the final output:

$$\mathcal{O} = \boldsymbol{x}\boldsymbol{W}_0 + \mathcal{O}^{\text{H-LoRA}}. \quad (11)$$

### 4.4. Training Pipeline

**1st Stage: multimodal Alignment.** In the first stage, we design separate visual adapters and H-LoRA submodules for medical unified tasks. For the medical comprehension

Table 1: Comparison of HealthGPT with other LVLMs and unified multimodal models on medical visual comprehension tasks. **Bold** and underlined text indicates the best performance and second-best performance, respectively.

| Type | Model | # Params | Medical LVLM | VQA-RAD ↑ close | VQA-RAD ↑ all | SLAKE ↑ close | SLAKE ↑ all | PathVQA ↑ close | PathVQA ↑ all | MMMU -Med | OMVQA↑ | Avg. ↑ |
|---|---|---|---|---|---|---|---|---|---|---|---|---|
| **Comp. Only** | Med-Flamingo | 8.3B | ✓ | 58.6 | 43.0 | 47.0 | 25.5 | 61.9 | 31.3 | 28.7 | 34.9 | 41.4 |
| | LLaVA-Med | 7B | ✓ | 60.2 | 48.1 | 58.4 | 44.8 | 62.3 | 35.7 | 30.0 | 41.3 | 47.6 |
| | HuatuoGPT-Vision | 7B | ✓ | 66.9 | 53.0 | 59.8 | 49.1 | 52.9 | 32.0 | 42.0 | 50.0 | 50.7 |
| | BLIP-2 | 6.7B | ✗ | 43.4 | 36.8 | 41.6 | 35.3 | 48.5 | 28.8 | 27.3 | 26.9 | 36.1 |
| | LLaVA-v1.5 | 7B | ✗ | 51.8 | 42.8 | 37.1 | 37.7 | 53.5 | 31.4 | 32.7 | 44.7 | 41.5 |
| | InstructBLIP | 7B | ✗ | 61.0 | 44.8 | 66.8 | 43.3 | 56.0 | 32.3 | 25.3 | 29.0 | 44.8 |
| | Yi-VL | 6B | ✗ | 52.6 | 42.1 | 52.4 | 38.4 | 54.9 | 30.9 | 38.0 | 50.2 | 44.9 |
| | InternVL2 | 8B | ✗ | 64.9 | 49.0 | 66.6 | 50.1 | 60.0 | 31.9 | 43.3 | 54.5 | 52.5 |
| | Llama-3.2 | 11B | ✗ | 68.9 | 45.5 | 72.4 | 52.1 | 62.8 | 33.6 | 39.3 | 63.2 | 54.7 |
| **Comp. & Gen.** | Show-o | 1.3B | ✗ | 50.6 | 33.9 | 31.5 | 17.9 | 52.9 | 28.2 | 22.7 | 45.7 | 42.6 |
| | Unified-IO 2 | 7B | ✗ | 46.2 | 32.6 | 35.9 | 21.9 | 52.5 | 27.0 | 25.3 | 33.0 | 33.8 |
| | Janus | 1.3B | ✗ | 70.9 | 52.8 | 34.7 | 26.9 | 51.9 | 27.9 | 30.0 | 26.8 | 33.5 |
| | Janus-Pro | 7B | ✗ | 62.9 | 49.2 | 51.3 | 38.7 | 48.9 | 27.2 | 32.7 | 54.1 | 45.6 |
| | Emu3 | 8B | ✗ | 72.1 | 55.2 | 57.3 | 39.8 | 54.1 | 30.6 | 28.7 | 39.5 | 47.2 |
| | HealthGPT-M3 | 3.8B | ✓ | 73.7 | 55.9 | 74.6 | 56.4 | 78.7 | 39.7 | 43.3 | 68.5 | 61.3 |
| | HealthGPT-L14 | 14B | ✓ | 77.7 | 58.3 | 76.4 | 64.5 | 85.9 | 44.4 | 49.2 | 74.4 | 66.4 |
| | HealthGPT-XL32 | 32B | ✓ | **78.1** | **60.5** | **83.7** | **68.2** | **92.4** | **50.9** | **58.0** | **77.2** | **71.1** |

task, we train abstract-grained visual adapters using high-quality image-text pairs to align visual embeddings with textual embeddings, thereby enabling the model to accurately describe medical visual content. During this process, the pre-trained LLM and its corresponding H-LoRA submodules remain frozen. In contrast, the medical generation task requires training concrete-grained adapters and H-LoRA submodules while keeping the LLM frozen. Meanwhile, we extend the textual vocabulary to include multimodal tokens, enabling the support of additional VQGAN vector quantization indices. The model trains on image-VQ pairs, endowing the pre-trained LLM with the capability for image reconstruction. This design ensures pixel-level consistency of pre- and post-LVLM. The processes establish the initial alignment between the LLM's outputs and the visual inputs.

**2nd Stage: Heterogeneous H-LoRA Plugin Adaptation.** The submodules of H-LoRA share the word embedding layer and output head but may encounter issues such as bias and scale inconsistencies during training across different tasks. To ensure that the multiple H-LoRA plugins seamlessly interface with the LLMs and form a unified base, we fine-tune the word embedding layer and output head using a small amount of mixed data to maintain consistency in the model weights. Specifically, during this stage, all H-LoRA submodules for different tasks are kept frozen, with only the word embedding layer and output head being optimized. Through this stage, the model accumulates foundational knowledge for unified tasks by adapting H-LoRA plugins.

**3rd Stage: Visual Instruction Fine-Tuning.** In the third stage, we introduce additional task-specific data to fur-

ther optimize the model and enhance its adaptability to downstream tasks such as medical visual comprehension (e.g., medical QA, medical dialogues, and report generation) or generation tasks (e.g., super-resolution, denoising, and modality conversion). Notably, by this stage, the word embedding layer and output head have been fine-tuned, only the H-LoRA modules and adapter modules need to be trained. This strategy significantly improves the model's adaptability and flexibility across different tasks.

## 5. Experiments

### 5.1. Data and Experimental Setup

**Data Details.** We curate **VL-Health** dataset (see Figure 4). For medical visual comprehension, we leverage multiple medical-specific datasets, including PubMedVision (Chen et al., 2024a), LLaVA-Med (Li et al., 2024a), PathVQA (He et al., 2020), MIMIC-CXR-VQA (Bae et al., 2024), SLAKE (Liu et al., 2021), and VQA-RAD (Lau et al., 2018). Additionally, we incorporate high-quality open-world data from LLaVA-1.5 (Liu et al., 2024b) to preserve the model's general knowledge and instruction-following capabilities. For generation tasks, we construct a reconstruction dataset based on LLaVA-558k (Liu et al., 2024b), and also explore two key tasks in personalized medical image enhancement—super-resolution and modality conversion—using the IXI (Davies et al., 2014) and SynthRAD2023 (Thummerer et al., 2023) datasets. In addition to the above datasets, we further evaluate the comprehension task on the OmniMedVQA (Hu et al., 2024) and MMMU (Yue et al., 2024) benchmarks. Detailed data

Table 2: The experimental results for the four modality conversion tasks.

| Model | CT to MRI (Brain) | | | CT to MRI (Pelvis) | | | MRI to CT (Brain) | | | MRI to CT (Pelvis) | | |
|---|---|---|---|---|---|---|---|---|---|---|---|---|
| | SSIM ↑ | PSNR ↑ | MSE ↓ | SSIM ↑ | PSNR ↑ | MSE ↓ | SSIM ↑ | PSNR ↑ | MSE ↓ | SSIM ↑ | PSNR ↑ | MSE ↓ |
| pix2pix | 71.09 | 32.65 | 36.85 | 59.17 | 31.02 | 51.91 | 78.79 | 33.85 | 28.33 | 72.31 | 32.98 | 36.19 |
| CycleGAN | 54.76 | 32.23 | 40.56 | 54.54 | 30.77 | 55.00 | 63.75 | 31.02 | 52.78 | 50.54 | 29.89 | 67.78 |
| BBDM | 71.69 | 32.91 | 34.44 | 57.37 | 31.37 | 48.06 | **86.40** | 34.12 | 26.61 | 79.26 | 33.15 | 33.60 |
| Vmanba | 69.54 | 32.67 | 36.42 | 63.01 | 31.47 | 46.99 | 79.63 | 34.12 | 26.49 | 77.45 | 33.53 | 31.85 |
| DiffMa | 71.47 | 32.74 | 35.77 | 62.56 | 31.43 | 47.38 | 79.00 | 34.13 | 26.45 | 78.53 | 33.68 | 30.51 |
| HealthGPT-M3 | 79.38 | 33.03 | 33.48 | 71.81 | 31.83 | 43.45 | 85.06 | **34.40** | **25.49** | 84.23 | **34.29** | **27.99** |
| HealthGPT-L14 | **79.73** | **33.10** | **32.96** | **71.92** | **31.87** | **43.09** | 85.31 | 34.29 | 26.20 | **84.96** | 34.14 | 28.13 |

Table 3: Comparison results of super-resolution task.

| Model | SSIM ↑ | PSNR ↑ | MSE ↓ | LPIPS ↓ |
|---|---|---|---|---|
| SRGAN | 71.34 | 32.01 | 41.27 | 24.50 |
| DASR | 71.57 | 32.34 | 38.25 | 19.17 |
| Real-ESRGAN | 67.30 | 31.87 | 42.57 | 20.64 |
| LIIF | 73.27 | 32.13 | 40.14 | 22.93 |
| BSRGAN | 69.97 | 31.97 | 41.52 | 28.72 |
| HealthGPT-M3 | **78.19** | **32.76** | **34.47** | **12.02** |
| HealthGPT-L14 | 77.94 | 32.71 | 35.19 | 12.43 |

selection and instruction templates are in the Appendix A.3.

**Model Details.** We select CLIP-L/14 (Radford et al., 2021) as the visual encoder and used the hidden states of its second and penultimate layers as concrete-grained and abstract-grained features for dynamic hierarchical visual perception of models. Drawing on the successful experiences of LLaVA architectures, we employ a MLP to align the multimodal feature embeddings. We choose the parameter-efficient phi-3-mini (Abdin et al., 2024) and phi-4 (Abdin et al., 2024) as the base models. Additionally, we employ the larger-scale Qwen2.5 (Team, 2024b) to explore the scalability of the proposed method. For visual comprehension and generation tasks, we set the rank of H-LoRA to $16/8$ and $64/32$ for different base models, with four experts. Additionally, we use the f8-8192 version of VQGAN as the image indexing and upsampling module. The model parameters and training configurations are provided in Appendix A.1 and A.2.

**5.2. Main Experiments**

**Comprehension.** We compare HealthGPT with several existing models, including medical-specific LVLMs (e.g., Med-Flamingo (Moor et al., 2023), LLaVA-Med (Li et al., 2024a), HuatuoGPT-Vision (Chen et al., 2024a)) as well as recent open-world LVLMs (e.g., BLIP-2 (Li et al., 2023b), LLaVA-v1.5 (Liu et al., 2024b), InstructBLIP (Dai et al., 2023), Yi-VL (Young et al., 2024), InternVL2 (Chen et al., 2024b), Llama-3.2 (Dubey et al., 2024)). Additionally, we test several SOTA unified visual comprehension and generation models, including Show-o (Xie et al., 2024), Unified-IO 2 (Lu et al., 2024), Janus (Wu et al., 2024a), Janus-Pro (Chen

et al., 2025), and Emu3 (Wang et al., 2024a). The experimental results are shown in Table 1, with the following key observations: **(i) SOTA Results Compared with LVLMs:** In medical visual comprehension tasks, HealthGPT demonstrates superior performance, significantly outperforming both medical-specific models (e.g., HuatuoGPT-Vision) and general-purpose models (e.g., Llama-3.2). **(ii) Surpassing Current Unified LVLMs:** Despite being trained on billions of data points, unified models still exhibit poor generalization performance in medical visual comprehension. For instance, Unified-IO 2 scored only 33.8. In contrast, *HealthGPT-M3*, with only 3.8B parameters, scored 61.3 on the medical multimodal unified task, significantly outperforming existing unified models in medical downstream scenarios. **(iii) Stable Improvement with Large Base Model:** Our method demonstrates excellent scalability, with *HealthGPT-L14* and *HealthGPT-XL32* achieving a score of 66.4 and 71.1 in the larger model configuration. This result significantly outperforms all other models, highlighting the effectiveness of scaling up the base model for enhanced performance in medical tasks.

It is noteworthy that some Med-LVLMs (Xie et al., 2025) perform continual fine-tuning and evaluation on the training sets of specific benchmarks, which may lead to structural degradation in model outputs and negatively impact generalization. To ensure fair comparison and reliable evaluation of generalization ability, we adopt a unified experimental setting and only compare with models trained on diverse and comprehensive datasets.

**Generation.** We study three key tasks in medical imaging. **(i) Modality Conversion:** In this task, we focus on the conversion between CT and MRI modalities for the brain and pelvic regions, designing four specific sub-tasks. All comparative models (Pix2Pix (Isola et al., 2017), CycleGAN (Zhu et al., 2017), BBDM (Li et al., 2023a), Vmamba (Liu et al., 2024e), and DiffMa (Wang et al., 2024b)) trained a separate model for each sub-task, while HealthGPT unify all tasks into a single training process. The experimental results, shown in Table 2, demonstrate that our approach outperforms other methods across multiple evaluation metrics. For instance, in the CT2MRI-Brain

Table 4: We present the performance and speed differences of LoRA, MoELoRA (n=4), and H-LoRA (n=4) on medical visual comprehension and generation tasks.

| Model | | VQA-RAD | | SLAKE | | Comp. PathVQA | | MMMU -Med | OMVQA | Gen. | | | Training Time |
|---|---|---|---|---|---|---|---|---|---|---|---|---|---|
| | | close | all | close | all | close | all | | | RECOM | MTRANS | SR | |
| HealthGPT w/ | +LoRA | 71.3 | **57.2** | 70.0 | 53.4 | 76.4 | 38.6 | 41.30 | 65.10 | 62.67 | 59.99 | 65.88 | **1.00×** |
| | +MoELoRA | 72.5 | **57.2** | 66.4 | 52.4 | 73.2 | 36.0 | 39.30 | 64.90 | 67.31 | 59.76 | 65.91 | 1.49× |
| | +H-LoRA | **73.7** | 55.9 | **74.6** | **56.4** | **78.7** | **39.7** | **43.30** | **68.50** | **67.69** | **60.30** | **66.14** | 1.00× |

Table 5: Comparison between the H-LoRA-based Three-Stage Learning Strategy and the mixed-training approach.

| Training Strategy | | VQA-RAD | | SLAKE | | Comp. PathVQA | | MMMU -Med | OMVQA | Gen. | | | |
|---|---|---|---|---|---|---|---|---|---|---|---|---|---|
| | | close | all | close | all | close | all | | | CT Brain | Pelvis | MRI Brain | Pelvis |
| HealthGPT w/ | Mixed-Training | 56.6 | 37.9 | 45.0 | 32.9 | 65.7 | 33.6 | **44.0** | 48.9 | 65.64 | 62.75 | 56.61 | 50.77 |
| | Three-Stage-Training | **72.5** | **55.2** | **77.9** | **59.6** | **79.7** | **49.0** | 42.7 | **68.5** | **70.84** | **72.99** | **65.26** | **61.33** |

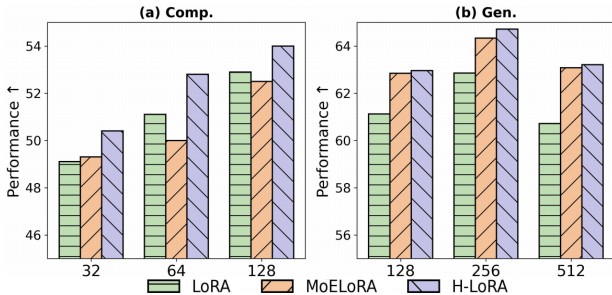

Figure 5: Performance comparison of LoRA, MoELoRA, and H-LoRA under different rank settings.

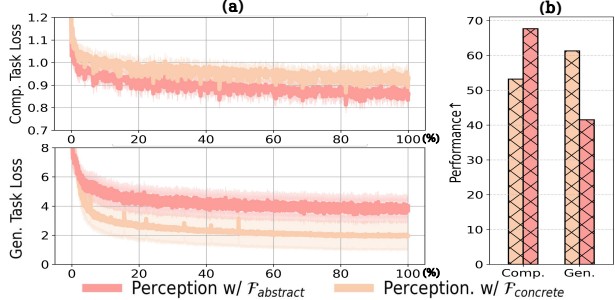

Figure 6: The loss visualization (a) and performance comparison (b) with respect to different visual perceptions.

task, *HealthGPT-M3* achieves an SSIM of 79.38, significantly surpassing traditional methods like Pix2Pix (71.09) and the recent DiffMa (71.47). **(ii) Super-Resolution:** We conduct 4× super-resolution experiments on the IXI dataset, with the results presented in Table 3. Notably, most existing methods fail to fully leverage the prior knowledge of key structures in medical images, resulting in significant shortcomings in detail recovery. In contrast, our method significantly mitigates this issue. Specifically, *HealthGPT-M3* excels in key metrics such as SSIM, PSNR, and MSE, achieving scores of 78.19, 32.76, and 34.47, respectively. Additionally, *HealthGPT-M3* achieves the lowest perception distance of 12.34, further validating its exceptional performance in human visual perception. **(iii) Reconstruction:** We compare *HealthGPT-M3* with unified models with reconstruction capabilities, such as Unified-IO 2 and SEED-X. The results show that our approach performs better controllability for visual reconstruction. Details are in the Appendix C.5.

To explore the scalability of the proposed method in generation tasks, we also train *HealthGPT-L14* with a similar number of trainable parameters to the *M3* version. Hence,

the similar performance between the two models in Figure 2 and 3 meets our expectations. For efficiency considerations, most open-world unified models are limited to a size of no more than 13B. In alignment with this practical constraint, we did not train the *HealthGPT-XL32* on generation tasks.

### 5.3. In-Depth Study

**Effect of Heterogeneous Low-Rank Adaptation.** H-LoRA provides an optimized multi-LoRA architecture for multi-task learning. We conduct extensive validation of this structure, with results presented in Table 4, comparing the performance of LoRA, MoELoRA, and H-LoRA in medical unified comprehension and generation tasks. In the majority of comprehension tasks and all generation tasks, H-LoRA demonstrates superior performance, particularly in the OmniMedVQA benchmark, where it improved from 64.90 to 68.50. Notably, despite some applications of MoELoRA in certain scenarios, it do not show advantages in this task and had a training time approximately 50% longer than LoRA. Figure 5 illustrates the performance of the three PEFT methods in medical visual comprehension and generation tasks

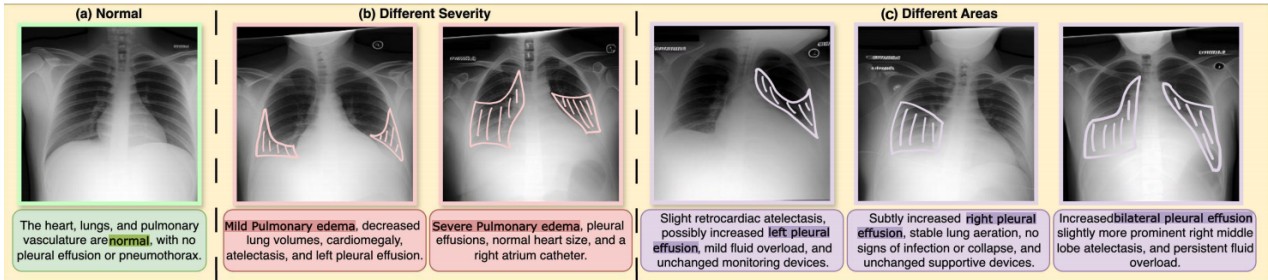

Figure 7: Case study of report-to-CXR under different instructions. (a) shows a normal CXR image for comparison. (b) and (c) illustrate generated cases with varying severity and affected regions. The graffiti areas indicate abnormal conditions.

across different ranks, with H-LoRA consistently outperforming the other methods in all scenarios, demonstrating significant advantages in handling diverse tasks.

**Different Learning Strategy.** We propose a three-stage learning strategy for H-LoRA that decouples comprehension and generation tasks. Unlike methods that train both tasks simultaneously, our approach reduces performance degradation from task conflicts (see Table 5). In the medical visual comprehension task, mixed training causes catastrophic forgetting and degrades visual reconstruction, whereas our strategy effectively uses the medical embedding knowledge in pre-trained LLMs to mitigate these conflicts. Meanwhile, we examine how fusing heterogeneous H-LoRA plugins in the second training stage results in minimal performance degradation. Detailed results are in the Appendix C.3.

**Hierarchical Visual Perception Analysis.** We conduct an ablation analysis on visual perceptual inputs suitable for comprehension and generation tasks. Figure 6 illustrates that the convergence efficiency for comprehension tasks is significantly higher with abstract-grained visual inputs compared to concrete-grained inputs, whereas generation tasks perform better with concrete-grained inputs. This result further underscores the necessity of the hierarchical visual perception we propose, which suggests that customizing visual inputs at different hierarchies for specific tasks can substantially enhance efficiency.

**Report-to-CXR Task.** Text-to-image generation is a common component in unified models (Lee et al., 2023; Huang et al., 2023; 2024); therefore, we further extend our investigation to this task within the medical domain. We explore the Report-to-CXR task without reference images, using a small amount of CXR data (Johnson et al., 2019) for instruction fine-tuning. Figure 7 annotates images with varying injury degrees and locations, comparing them to healthy CXR images. We observe that HealthGPT effectively generates CXR images based on the instructions, showcasing its potential in healthcare education and auxiliary diagnosis.

## 6. Conclusion

In this paper, we introduce HealthGPT, a Med-LVLM that unifies medical vision-language understanding and generation through a novel heterogeneous knowledge adaptation approach. Experimental results demonstrate that HealthGPT achieves significant performance improvements across multiple medical understanding and generation tasks, showcasing its potential for healthcare applications.

## Acknowledgements

This work has been supported in part by the NSFC (No. 62402426, 62272411), the Key Research and Development Projects in Zhejiang Province (No. 2025C01128, 2024C01106, 2025C01030, 2025C02156), Ningbo Yongjiang Talent Introduction Programme (2023A-400-G), Zhejiang University Education Foundation Qizhen Scholar Foundation.

## Impact Statement

The broader impact of HealthGPT, designed as a multimodal assistant for healthcare, holds the potential to significantly benefit a wide range of multimodal healthcare applications. Our research investigates the innovative, parameter-efficient fine-tuning approach, H-LoRA, which is combined with hierarchical visual perception and a three-stage learning strategy. This method effectively integrates diverse comprehension and generation knowledge into pre-trained large language models. Besides, the proposed VL-Health serves as a fundamental basis for investigating medical vision-language comprehension and generation. We are confident in its superiority as a state-of-the-art approach for medical multimodal comprehension and generation, and we anticipate its substantial influence on future research directions in the field.

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

# Appendix

This is the Appendix for "HealthGPT: A Medical Large Vision-Language Model for Unifying Comprehension and Generation via Heterogeneous Knowledge Adaptation". This Appendix is organized as follows:

- **Section A** presents the experimental implementation details, the training process of **HealthGPT**, and the specifics of **VL-Health**.

- **Section B** systematically provides an analysis of Heterogeneous Low-Rank Adaptation.

- **Section C** shows supplementary experimental results to validate the effectiveness of HealthGPT.

## A. Implementation Details

### A.1. Model Details

We employ CLIP-L/14 (Radford et al., 2021) as the visual feature extractor, extracting both shallow and deep features to serve as visual tokens. The model uses alignment adapters, implemented with two-layer MLPs, to align shallow features, representing concrete visual granularity, and deep features, representing abstract visual granularity. These visual tokens are concatenated with text tokens and input into the Large Language Models (LLMs).

The development of MLLMs has undergone several paradigm shifts (Zhang et al., 2024a; OpenAI, 2023; Liu et al., 2024c; Chen et al., 2024b); however, the LLaVA architecture (Liu et al., 2023) has gained widespread attention due to its simplicity and portability. Therefore, we adopt this architecture in our work. HealthGPT offers three versions: *HealthGPT-M3*, *HealthGPT-L14*, and *HealthGPT-XL32*, which are based on Phi-3-mini (Abdin et al., 2024), Phi-4 (Abdin et al., 2024), and Qwen2.5-32B (Team, 2024b) as the pre-trained base models, respectively. In addition, we expand the LLM vocabulary with 8192 VQ indices derived from VQGAN-f8-8192 (Esser et al., 2021), serving as multi-modal tokens to further augment the capacity of models for understanding both visual and textual input and output. Figure 6 shows the details.

Table 6: Overview of the Components of HealthGPT.

| Model | ViT | Adapter | MLP Dims | Model Dims | Base Model | Params | Vocab Size | H-LoRA Rank |
|-------|-----|---------|----------|------------|------------|--------|------------|-------------|
| *HealthGPT-M3* | CLIP-L/14 | 2-layer MLP | 1024 | 3072 | Phi-3-mini | 3.8B | 40206 | 16(Comp.), 64(Gen.) |
| *HealthGPT-L14* | CLIP-L/14 | 2-layer MLP | 1024 | 5120 | Phi-4 | 14B | 108547 | 8(Comp.), 32(Gen.) |
| *HealthGPT-XL32* | CLIP-L/14 | 2-layer MLP | 1024 | 5120 | Qwen2.5-32B | 32B | 152064 | 8(Comp.), 32(Gen.) |

### A.2. Training Details

In this study, we propose a three-stage learning strategy that is compatible with our innovative heterogeneous low-rank adaptation (H-LoRA). We provide a detailed hyperparameter configuration for the model's three-stage training process. The specific hyperparameter settings used are listed in Table 7. These hyperparameters are crucial for ensuring the model's learning efficacy and final performance.

Table 7: Overview of Hyperparameter Configurations.

| Hyperparameter | HealthGPT-M3 | | | | | | HealthGPT-L14/XL32 | | | | | |
|----------------|--------------|------|--------------|------|--------------|------|--------------------|------|--------------|------|--------------|------|
| | Stage-1 | | Stage-2 | | Stage-3 | | Stage-1 | | Stage-2 | | Stage-3 | |
| | Comp. | Gen. | Comp. | Gen. | Comp. | Gen. | Comp. | Gen. | Comp. | Gen. | Comp. | Gen. |
| Optimizer | AdamW | | AdamW | | AdamW | | AdamW | | AdamW | | AdamW | |
| Adapter LR | 1e-3 | 2e-5 | 2e-5 | | 2e-5 | | 1e-3 | 2e-5 | 2e-5 | | 2e-5 | |
| Learning Rate | / | 2e-4 | 2e-4 | | 2e-4 | | / | 1e-4 | 2e-4 | | 2e-4 | |
| Global Batch Size | 256 | 64 | 32 | | 128 | 64 | 256 | 64 | 32 | | 128 | 64 |
| Weight Decay | 0 | | 0 | | 0 | | 0 | | 0 | | 0 | |
| Dropout Rate | 0 | 0.05 | 0.05 | | 0.05 | | 0 | 0.05 | 0.05 | | 0.05 | |
| LR Scheduler | Warm Up | | Constant | | Warm Up | | Warm Up | | Constant | | Warm Up | |
| Max Sequence Length | 2048 | | 2048 | | 2048 | | 2048 | | 2048 | | 2048 | |

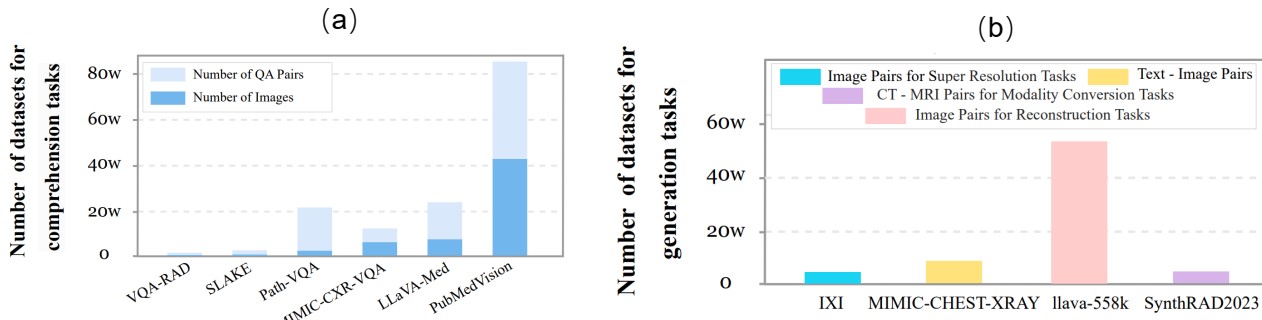

Figure 8: VL-Health dataset collection distribution.

It is worth noting that we sometimes observe instances of loss spikes during the training of medical visual comprehension and generation tasks. Through repeated validation, we discovered that larger model parameters and learning rates tend to lead to this issue, which is the reason for the slight differences in hyperparameters between *HealthGPT-M3* and *HealthGPT-L14/XL32*.

## A.3. VL-Health

The construction of the VL-Health dataset involves two key steps: **(i) data collection**, **(ii) data processing**, as detailed below:

**Data Collection:** During the collection phase, we carefully considered the diversity of medical images and the complexity of the tasks, selecting appropriate subsets for comprehension and generation tasks. For comprehension tasks, we selected datasets such as VQA-RAD (Lau et al., 2018), SLAKE (Liu et al., 2021), PathVQA (He et al., 2020), and MIMIC-CXR-VQA (Bae et al., 2024), which cover various medical imaging modalities like radiology and pathology, and include professional annotations to assist the model in learning tasks such as lesion detection and disease diagnosis. Additionally, large-scale multi-modal datasets like LLaVA-Med (Li et al., 2024a) and PubMedVision (Chen et al., 2024a) were included to provide broader medical knowledge support and facilitate the training of complex reasoning tasks. For generation tasks, we focused on four mainstream task categories: super-resolution image generation, modality conversion, text-to-image generation, and image reconstruction. The IXI (Davies et al., 2014) dataset, containing a large number of healthy brain MRI images, is suitable for training super-resolution models; the MIMIC-CHEST-XRAY (Bae et al., 2024) dataset, with X-ray images and their corresponding textual reports, is appropriate for text-to-image generation tasks; the SynthRAD2023 (Thummerer et al., 2023) dataset provides a large number of paired CT and MRI images, supporting modality conversion model training; for image reconstruction tasks, we rewrote and adjusted the LLaVA-558k (Liu et al., 2024b) dataset.

**Data Processing:** After data collection, we performed filtering and processing of the raw data. For VisualQA tasks, we standardized the data entries into two forms: open-ended questions and single-choice questions, enabling flexible training and evaluation. Additionally, considering that multi-image data has a minimal impact on performance but introduces extra padding and training time, we excluded multi-image data. For the scanned image data in generation tasks, we applied slicing extraction, image registration, data augmentation, and normalization to treat 2D images as visual inputs for model training or used VQGAN-generated indices to supervise the generation tasks.

### A.3.1. DATA STATISTICS

This section provides detailed statistical information about the VL-Health dataset to offer a more comprehensive understanding.

**Data Overview:** To ensure a balanced development of the model's comprehension and generation capabilities, in addition to the LLaVA-558k and PubMedVision-PT datasets used for alignment, the `VL-Health` dataset ultimately selected 765,802 additional visual question-answering (VQA) training samples (to endow the model with visual comprehension and instruction-following capabilities) and 783,045 generation training samples (to provide the model with reconstruction and visual generation instruction-following abilities). This contributes to the transfer of knowledge between comprehension and generation tasks, enhancing the model's overall performance. For medical image comprehension tasks, images were

selected from VQA-RAD (approximately 450 images), SLAKE (approximately 630 images), PathVQA (approximately 2,600 images), MIMIC-CXR-VQA (approximately 52,000 images), LLaVA-Med (approximately 61,000 images), and PubMedVision (approximately 500,000 images). Multiple question-answer pairs were retained for each image to enhance the model's understanding and generalization of the image content. Table 8 shows the data distribution of VL-Health for three-stage learning strategy, where mixed-47k is based on the sampling of all data in stage-1.

**Diversity and Quality Assessment:** VL-Health covers 11 modalities, including CT, MRI, X-ray, microscopy, OCT, ultrasound, and fundus photography, which aids the model in learning features from various modalities. The dataset also encompasses a wide range of diseases, from common to rare, and from localized lesions to systemic diseases, including pulmonary diseases, skeletal abnormalities, brain lesions, tumors, cardiovascular diseases, and cellular abnormalities. This provides comprehensive training support to the model, enabling it to learn the characteristics and diagnosis of various diseases.

Table 8: Data distribution of VL-Health in three-stage learning strategy.

| Medical Task | Stage-1 | Stage-2 |
|---|---|---|
| Comp. Gen. | LLaVA-558k, PubMedVision-PT LLaVA-558k | Mixed-47k |
| **Medical Task** | **Stage-3** | |
| Comp. Gen. | LLaVA_Med, MIMIC_CXR_VQA, PubMedVision-FT, LLaVA-665k, PathVQA, SLAKE, VQA-RAD IXI, SynthRAD2023, MIMIC-CHEST-XRAY | |

A.3.2. DATA FORMAT.

All data samples are converted into a unified instruction-response format for training and evaluation. Specifically, the VL-Health dataset consists of the following components:

- *Task Type*: Specifies the granularity of visual features output by the visual encoder and selects the corresponding H-LoRA submodule. For generation tasks, the response also includes multi-modal tokens corresponding to VQ indices.

- *Task Instruction*: Guides the model to interpret the image and generate a response, covering various aspects of the image and specifying the output format.

- *Response*: The textual output generated based on the task instruction and input image, ensuring it meets the question and formatting requirements.

- *Input Image*: Provides the visual signal for the model to process.

- *Target Image Index*: In generation tasks, this is added as a multi-modal token to the response for autoregressive generation.

## B. Analysis of Heterogeneous Low-Rank Adaptation

We propose H-LoRA, which utilizes hard routing selection to allocate plugins for knowledge learning and representation across tasks, thereby preventing conflicts arising from heterogeneous knowledge. This is especially beneficial for domain adaptation, where task and distribution shifts often cause interference between domains (Zhang et al., 2024b). Furthermore, within each task, we optimized based on MoELoRA, enhancing performance while reducing computational overhead. The pseudocode is detailed Algorithm 1.

We further analyzed the computational overhead differences between MoELoRA and H-LoRA. Assuming that both methods use the same number of LoRA experts $k$, we can compare their time complexity from the perspective of the operational steps involved.

**Computational Overhead of MoELoRA.** In MoELoRA, the operations involving the expert matrix mainly include the following steps: **(i) Expert Multiplication**: MoELoRA requires $2k$ multiplications with the LoRA experts. **(ii) Router Multiplication**: One multiplication with the Router is required. **(iii) Router Output Expansion**: MoELoRA needs to

---

**Algorithm 1** H-LoRA Algorithm

---

**Input:** concrete-grained visual features $\mathcal{F}^{\text{Con}}$, abstract-grained visual features $\mathcal{F}^{\text{Abs}}$, comprehension-based H-LoRA modules $(\{A_i^{\text{Comp.}}\}_{i=1}^k, \mathcal{R}_{\text{outer}}^{\text{Comp.}})$, generation-based H-LoRA modules $(\{A_i^{\text{Gen.}}\}_{i=1}^k, \mathcal{R}_{\text{outer}}^{\text{Gen.}})$, task type $T$ (comprehension or generation), number of LoRA experts $k$, origin linear layer weights $W_0$, text features $\mathcal{T}$, hidden state h

**Output:** final output $\mathcal{O}$

// Select task-specific image features

**if** $T = $ generation task **then**

  $\mathcal{F}^{\text{img}} \leftarrow \mathcal{F}^{\text{Con}}$

**else if** $T = $ comprehension task **then**

  $\mathcal{F}^{\text{img}} \leftarrow \mathcal{F}^{\text{Abs}}$

**end if**

$\mathcal{U} \leftarrow \text{concat}(\mathcal{F}^{\text{img}}, \mathcal{T})$ // Concatenate image features and text features

$\{A_i\}_{i=1}^k, \{B_i\}_{i=1}^k, \mathcal{R}_{\text{outer}} \leftarrow \{A_i^T\}_{i=1}^k, \{B_i^T\}_{i=1}^k, \mathcal{R}_{\text{outer}}^T$ // Assign task-specific H-LoRA submodule

// Merge LoRA experts' matrices

$\mathbf{A}^{\text{merged}} \leftarrow \text{concat}(\{A_i\}_{i=1}^k)$

$\mathbf{B}^{\text{merged}} \leftarrow \text{concat}(\{B_i\}_{i=1}^k)$

$\mathcal{W} \leftarrow R(\mathbf{h})$ // Generate routing weights based on input hidden state $\mathbf{x}$

$\mathcal{W}^{\text{expanded}} \leftarrow \alpha \times \mathcal{W}/r \otimes \mathbf{1}_r$ // Expand routing weights to match merged matrices

$\mathcal{O}^{\text{H-LoRA}} \leftarrow (x \cdot \mathbf{A}^{\text{merged}} \odot \mathcal{W}^{\text{expanded}}) \cdot \mathbf{B}^{\text{merged}}$ // Compute H-LoRA output using element-wise multiplication

$\mathcal{O} \leftarrow x \cdot W_0 + \mathcal{O}^{\text{H-LoRA}}$ // Add H-LoRA output to pre-trained weights to get final output

**Return** $\mathcal{O}$

---

perform $k$ expansion operations on the Router's output weights to generate the appropriate shapes that match the dimensions of the input and LoRA experts while iterating through the experts. **(iv) Dot Product**: For each expanded Router weight, a dot product with the intermediate state of the expert is required, resulting in $k$ multiplications. **(v) Addition**: Finally, $k$ addition operations are required to accumulate the results from each LoRA expert into the final output. Assuming the time complexity of each operation is the same, the additional time complexity introduced when equipping a fully connected layer with MoELoRA is: $O(2k + 1 + k + k + k) = O(5k + 1)$. Thus, MoELoRA introduces an additional time overhead of $O(5k + 1)$ during computation.

**H-LoRA.** In contrast to MoELoRA, H-LoRA reduces the computational overhead by concatenating the LoRA expert matrices. Specifically: **(i) Expert Multiplication**: H-LoRA merges all LoRA experts by directly creating a larger A and B matrix, instead of performing independent operations for each expert. This process can be implemented through matrix initialization without additional concatenation operations. Therefore, only 2 multiplications with the LoRA experts are required. **(ii) Router Multiplication**: H-LoRA still requires one multiplication with the Router. **(iii) Router Output Expansion**: H-LoRA only requires one expansion operation on the Router's output weights. **(iv) Dot Product**: H-LoRA only requires one dot product between the Router's output and the expert's intermediate state. **(v) Addition**: Finally, H-LoRA only requires one addition operation to accumulate the LoRA expert results into the intermediate state. Therefore, the additional time complexity introduced by H-LoRA is: $O(2 + 1 + 1 + 1 + 1) = O(6)$.

Comparing the two, we see that MoELoRA introduces a linear increase in additional time complexity with respect to the number of experts $k$, resulting in a complexity of $O(5k + 1)$, while H-LoRA's additional time complexity is fixed at $O(6)$, independent of $k$. We observe that when $k$ is small, the time complexity differences between MoELoRA and H-LoRA are negligible. However, as $k$ increases, MoELoRA's computational overhead grows linearly, while H-LoRA's remains constant. This makes H-LoRA significantly more computationally efficient than MoELoRA, particularly in large-scale tasks. We will further demonstrate the significant advantage of H-LoRA in training time in subsequent experiments, validating its efficiency in practical applications.

## C. Supplemental Experimental Results

In this section, we include additional experiments to demonstrate the superiority of HealthGPT and articulate our design philosophy.

## C.1. OmniMedVQA Benchmark

OmniMedVQA (Hu et al., 2024) is a novel, large-scale medical visual question answering (VQA) benchmark designed to encompass various modalities and anatomical regions by collecting diverse images from multiple medical datasets. Our experimental results are presented in Table 9.

Table 9: Performance comparison of OmniMedVQA Benchmark.

| Type | Model | Params | Medical LVLM | OmniMedVQA ↑ | | | | | | | |
|---|---|---|---|---|---|---|---|---|---|---|---|
| | | | | CT | X-ray | FDM | MiS | OCT | MRI | USS | Avg. |
| **Comp. Only** | Med-Flamingo | 8.3B | ✓ | 30.1 | 33.9 | 25.5 | 37.0 | 60.0 | 27.6 | 30.4 | 34.9 |
| | LLaVA-Med | 7B | ✓ | 28.4 | 32.8 | 42.7 | 31.6 | 55.3 | 45.0 | 53.6 | 41.3 |
| | HuatuoGPT-Vision | 7B | ✓ | 35.3 | 41.5 | 51.4 | 62.3 | 59.3 | 40.4 | 60.1 | 50.0 |
| | BLIP-2 | 6.7B | ✗ | 26.6 | 29.1 | 22.3 | 36.9 | 29.1 | 22.7 | 21.4 | 26.9 |
| | LLaVA-v1.5 | 7B | ✗ | 28.0 | 55.7 | 35.5 | 42.1 | 49.2 | 52.9 | 49.7 | 44.7 |
| | InstructBLIP | 7B | ✗ | 20.1 | 22.2 | 34.1 | 30.6 | 38.6 | 31.9 | 25.5 | 29.0 |
| | Yi-VL | 6B | ✗ | 51.2 | 47.1 | 27.7 | 62.6 | 67.6 | 55.0 | 40.3 | 50.2 |
| | InternVL2 | 8B | ✗ | 40.2 | 57.9 | 53.2 | 64.0 | 59.1 | 58.1 | 49.1 | 54.5 |
| | Llama-3.2 | 11B | ✗ | 37.6 | 55.2 | **71.4** | 82.1 | 62.5 | 65.2 | **68.6** | 63.2 |
| **Comp. & Gen.** | Show-o | 1.3B | ✗ | 29.0 | 50.4 | 30.9 | 22.0 | 30.8 | 34.2 | 33.8 | 33.0 |
| | Unified-IO 2 | 7B | ✗ | 10.8 | 37.7 | 12.3 | 25.3 | 32.6 | 30.9 | 37.7 | 26.8 |
| | Janus | 1.3B | ✗ | 24.9 | 54.8 | 35.9 | 62.7 | 54.2 | 50.7 | 36.8 | 45.7 |
| | Janus-Pro | 7B | ✗ | 31.2 | 61.2 | 51.4 | 71.7 | 60.2 | 57.2 | 45.8 | 54.1 |
| | Emu3 | 8B | ✗ | 34.0 | 31.2 | 34.5 | 34.8 | 78.3 | 51.7 | 12.3 | 39.5 |
| | HealthGPT-M3 | 3.8B | ✓ | 35.3 | 81.9 | 54.6 | 88.2 | 89.3 | 78.5 | 51.4 | 68.5 |
| | HealthGPT-L14 | 14B | ✓ | 39.0 | **86.6** | 64.1 | **88.6** | **99.7** | **80.9** | 62.2 | 74.4 |
| | HealthGPT-XL32 | 32B | ✓ | **66.4** | 81.3 | 67.7 | 85.6 | 98.7 | 76.3 | 64.5 | **77.2** |

Through our analysis, we make the following observations: (i) *HealthGPT-M3* outperforms other models in 4 out of 7 sub-tasks, achieving an average score that exceeds cutting-edge medical Large Vision-Language Models (Med-LVLMs) as well as general LVLMs; (ii) the unified model demonstrates relatively weak performance on OmniMedVQA; however, our approach effectively mitigates performance degradation caused by generation tasks, serving as a unified model; (iii) *HealthGPT-L14* and *HealthGPT-XL32* excel across all sub-tasks, achieving optimal or near-optimal results with an average score of 74.4 and 77.2, significantly surpassing other models.

## C.2. Stability Analysis of Number of Experts

We investigated the impact of the number of LoRA experts on model performance within a multi-LoRA architecture, conducting extensive experiments on MoELoRA and H-LoRA with varying numbers of experts. The experimental results are presented in Table 10. As the number of experts increases, the training time for MoELoRA is significantly prolonged. When $n = 8$, the training time for MoELoRA is twice that of LoRA, whereas H-LoRA incurs no additional training delay and performs better. It is estimated that at $n = 32$, the training time for MoELoRA could reach eight times that of LoRA, preventing it from completing training and inference. This result aligns with the analysis in Appendix B, indicating that H-LoRA not only avoids introducing additional training delays compared to LoRA but also outperforms MoELoRA.

Table 10: We explored the performance of MoELoRA and H-LoRA with different numbers of LoRA experts. At $n = 32$, MoELoRA was unable to complete training.

| Model | | n=2 | | | n=4 | | | n=8 | | | n=32 | | |
|---|---|---|---|---|---|---|---|---|---|---|---|---|---|
| | | Comp. | Gen. | Time | Comp. | Gen. | Time | Comp. | Gen. | Time | Comp. | Gen. | Time |
| HealthGPT w/ | +MoELoRA | 50.3 | 62.98 | 1.22× | 50.0 | 64.33 | 1.49× | 50.8 | 63.71 | 2.09× | / | / | 5.81× |
| | +H-LoRA | **51.5** | **63.48** | **0.99×** | **52.8** | **64.71** | **1.00×** | **53.6** | **64.98** | **0.99×** | **53.5** | **64.74** | **1.01×** |

### C.3. Impact of Heterogeneous Knowledge Fusion on Performance

Traditional unified models often utilize mixed training methods, which may result in performance degradation due to variations in task modes. To address this, we propose a three-phase learning strategy to support H-LoRA, effectively mitigating inter-task conflicts. Specifically, the second phase (Heterogeneous H-LoRA Plugin Adaptation) integrates LLMs with different H-LoRA plugins into a new unified foundation by mixing the training of the embedding layers and output heads for two tasks. Figure 9 illustrates the impact of this phase on the performance of medical comprehension and generation tasks. We observe that the second phase effectively unifies the model with minimal impact on overall performance, significantly alleviating the conflict issues arising from mixed training in medical scenarios.

### C.4. Human Evaluation.

We further conduct human evaluation on the VQA-RAD, SLAKE, and PathVQA benchmarks, which contain 1,000 open-ended questions. Specifically, we recruit 5 clinicians to rank the randomly shuffled responses from *HealthGPT-L14*, LLaVA-Med, HuatuoGPT-Vision, Llama-3.2, InternVL-2 and Show-o. During the evaluation, questions were randomly selected, and the model-generated responses were anonymized and ranked. The results, as shown in Figure 10, indicate that HealthGPT was frequently selected as the best answer. This suggests that HealthGPT has further application potential in medical care scenarios.

### C.5. Reconstruction Performance

Currently, unified models that align visual features based on reconstruction tasks include pre-LVLMs, post-LVLMs, as well as Unified-IO 2 (Lu et al., 2024) and SEED-X (Ge et al., 2024). To investigate the controllability of visual generation in rigorous settings such as medical contexts, we evaluated the performance of these models in medical image reconstruction in Table 11. Experimental results demonstrate that HealthGPT exhibits the most stable reconstruction performance with a small amount of data.

### C.6. Case Study

Figures 11 and 12 illustrate examples of modality transformation and super-resolution reconstruction. In Figure 11, the results generated by our method in the CT (MRI) to MRI (CT) transformation task are highly close to the ground truth, effectively guiding the model in the transformation across different regions. For the MRI super-resolution reconstruction task, Figure 12 demonstrates the accuracy of our method in restoring scan image details, accurately reconstructing the essential details of the image.

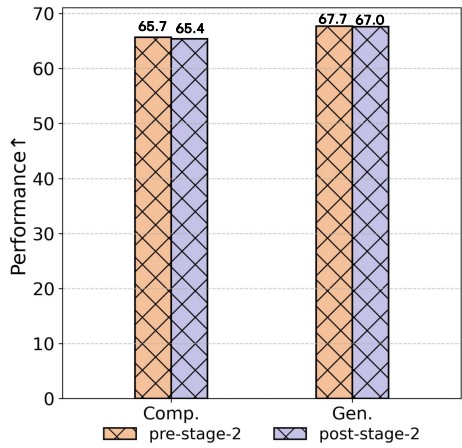

Figure 9: Performance changes before and after the stage-2.

Table 11: The experimental results for the four reconstruction tasks.

| Model | CT(Brain) | | | CT(Pelvis) | | | MRI (Brain) | | | MRI(Pelvis) | | |
|---|---|---|---|---|---|---|---|---|---|---|---|---|
| | SSIM ↑ | PSNR ↑ | MSE ↓ | SSIM ↑ | PSNR ↑ | MSE ↓ | SSIM ↑ | PSNR ↑ | MSE ↓ | SSIM ↑ | PSNR ↑ | MSE ↓ |
| SEED-X | 20.18 | 27.66 | 112.11 | 21.53 | 28.02 | 102.87 | 4.90 | 27.62 | 112.86 | 6.31 | 27.89 | 106.21 |
| Unified-IO 2 | 83.93 | 36.09 | 17.95 | 85.36 | 35.10 | 25.46 | 87.50 | **34.25** | **25.47** | **86.31** | **33.53** | **29.80** |
| HealthGPT-M3 | **91.73** | **36.42** | **15.46** | **94.26** | **37.30** | **12.53** | **88.76** | 33.97 | 27.05 | 84.40 | 33.11 | 32.62 |

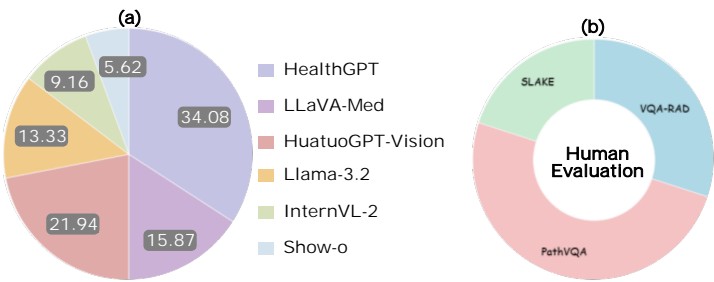

Figure 10: (a) Proportion of model responses selected as the best in human evaluation. (b) Human Evaluation Dataset.

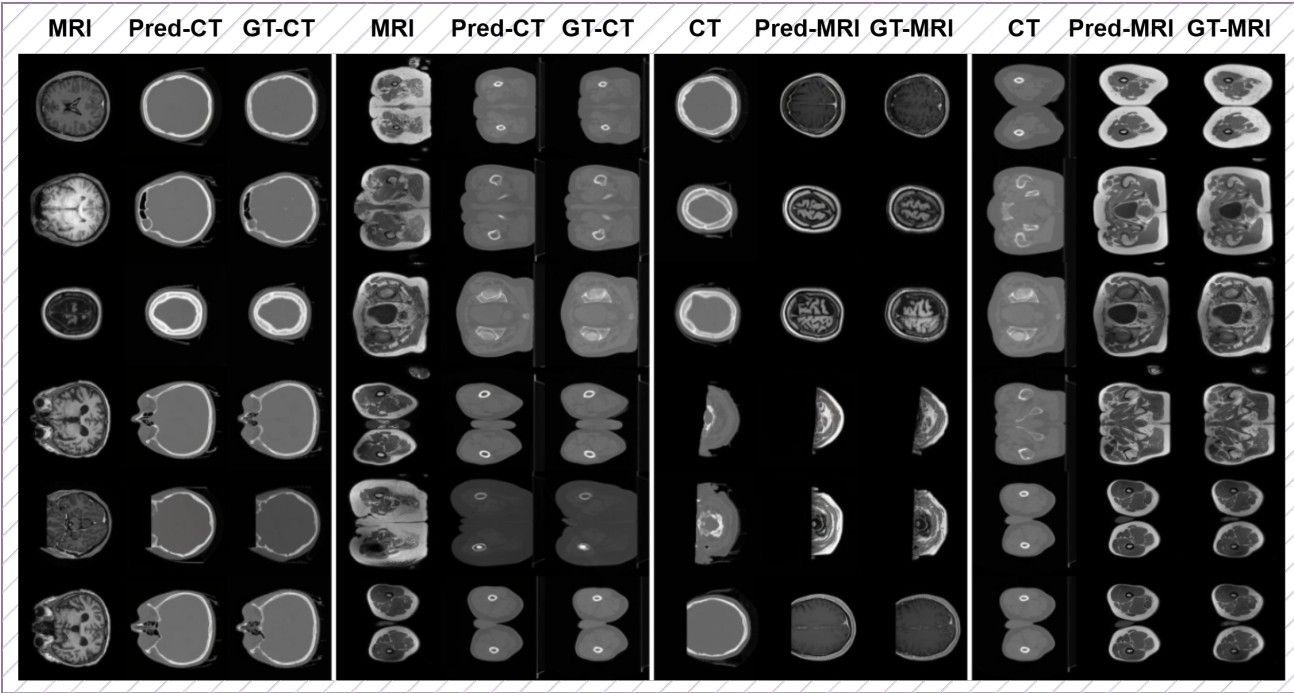

Figure 11: Case of modality transfer.

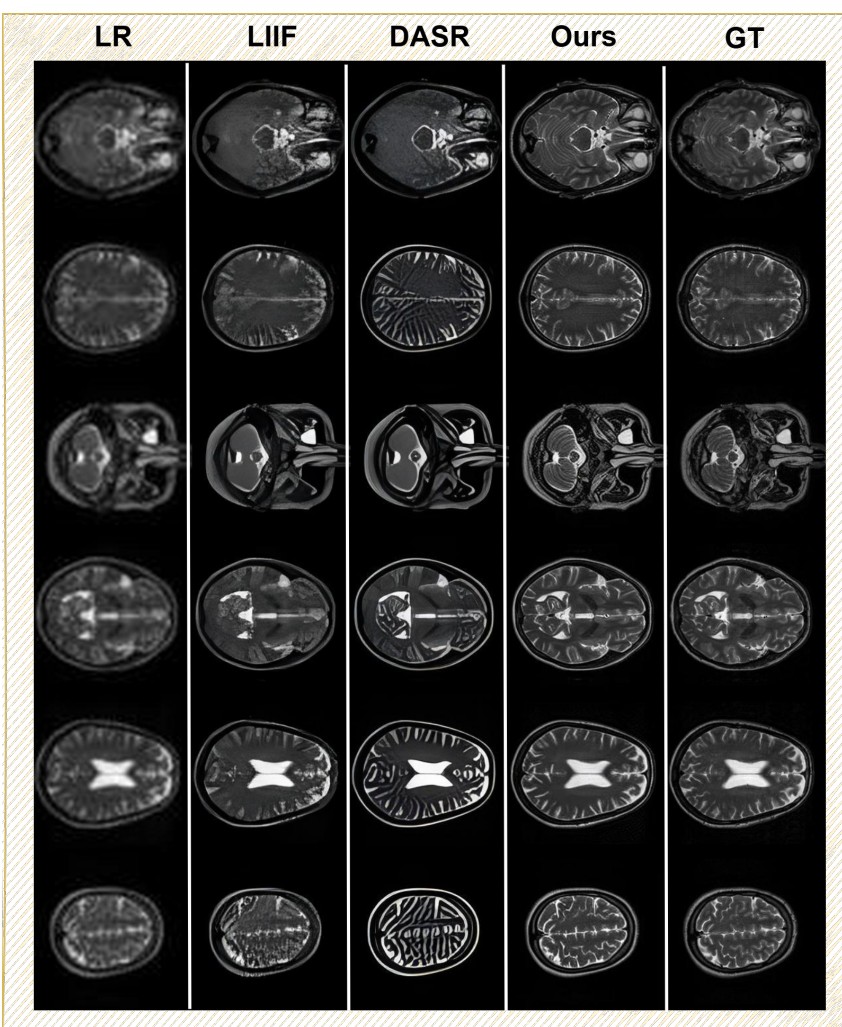

Figure 12: Case of MRI image super-resolution.

