# OpenReview forum: "HealthGPT: A Medical Large Vision-Language Model for Unifying Comprehension and Generation via Heterogeneous Knowledge Adaptation"
_ICML.cc/2025/Conference — ICML 2025 spotlightposter_

### Official Review · Reviewer_B8gX · 2025-02-18

**Overall Recommendation:** 4

**Summary:**

The paper introduces HealthGPT, a medical vision-language model that unifies visual comprehension and generation through heterogeneous knowledge adaptation. Key contributions include H-LoRA, a parameter-efficient fine-tuning method that decouples task-specific knowledge via independent low-rank plugins, a hierarchical visual perception strategy to handle abstract and concrete visual features, and the VL-Health dataset for medical multi-modal tasks. Experiments demonstrate HealthGPT outperforms state-of-the-art models in tasks like medical visual QA, image reconstruction, and modality conversion.

**Claims And Evidence:**

- In the Introduction, the authors claim there are conflicts between comprehension and generation and refer to Figure 2. What is the detailed experimental setting of this figure? Also, as mentioned by the authors, e.g. MetaMorph (Tong et al.) finds these two tasks are mutually beneficial. What do authors mean by "improvements still exhibit diminishing returns, with performance degradation remaining a significant issue" - line 71/72 (right col.)? I did not find corresponding evidence from this paper.
- Other claims are supported by experiments.

**Essential References Not Discussed:**

To my knowledge, most essential references are discussed. I found this paper relevant, can authors discuss and compare the performance with it: MedMax: Mixed-Modal Instruction Tuning for Training Biomedical Assistants. https://arxiv.org/abs/2412.12661

**Experimental Designs Or Analyses:**

- The authors have compared a series of models for medical visual comprehension. Can authors also compare with some stronger, state-of-the-art models such as GPT-4o and Qwen2-VL?
- There are analysis and ablation studies in section 5.3, which are helpful for understanding the proposed method.

**Methods And Evaluation Criteria:**

- There is no explanation of the fusion embedding layer in Figure 3, except a brief mention in line 120. What exactly is the fusion embedding layer? From Figure 3, it is after the text tokenizer and VQ tokenizer, so is it just the regular embedding layer for the conversion of the tokens?
- The evaluations are extensive and demonstrated the effectiveness of the proposed method.

**Other Comments Or Suggestions:**

NA

**Other Strengths And Weaknesses:**

###Strengths:

- The proposed method and dataset are meaningful explorations towards unified medical vision-language models. The proposed H-LoRA can efficiently enable the training.
- Experimental results in both comprehension and generation benchmarks indicate the effectiveness of the proposed model.

### Weaknesses:
Please address comments in other sections.

**Questions For Authors:**

- A pre-trained VQGAN is used for generation. Does the natural image pre-trained VQGAN work well for medical images?
- What is the total training time and GPUs required?
- Will the model weights be open-sourced? It's "coming soon" in the repo.

**Relation To Broader Scientific Literature:**

It is related to the general unified vision-language models (e.g. unified-IO, Janus) but specifically focuses on medical applications.

**Theoretical Claims:**

No theoretical claim.

---

> ### Author Rebuttal · Authors · 2025-03-30
>
> Thank you for your positive and insightful review, which greatly helps to refine the manuscript's quality and clarity. Below are our point-by-point responses.
>
> ## 1. Additional Details
>
> We appreciate the reviewer’s insightful questions and welcome the opportunity to further elaborate on the relevant concepts. Below, we provide detailed clarifications.
>
> **1.1 Fig.2 setting**
>
> Regarding the experimental setup in Figure 2, its purpose is to illustrate conflicts that arise in traditional joint modeling methods when comprehension and generation tasks are trained on heterogeneous data. The experiment consists of two stages:
> 1. In the first stage, lightweight adapters align visual and textual features to build the visual-language pretraining base.
> 2. In the second stage, while preserving single-task data integrity, we gradually introduce heterogeneous data from the other task (25%/50%/100% generation or comprehension data).
>
> We observe that increasing heterogeneous task data significantly degraded performance due to task interference. In contrast, our method introduces only 2% additional repeated data (used in the second phase of TLS), **greatly improving the performance of both tasks**.
>
>
> **1.2 Diminishing returns clarified**
>
> We would like to clarify a possible misunderstanding of the “diminishing returns” description. Our intent was not to undervalue methods like MetaMorph, but to highlight the difficulty of transferring joint training paradigms across tasks with distinct structures and imbalanced data scales.
>
> In the medical domain, comprehension and generation differ significantly in objectives and data availability, making mutual gains through joint training challenging and sometimes prone to negative transfer (see Fig. 2). We will revise the phrasing in the next version to avoid broad descriptions:
> > Joint training of comprehension and generation may face challenges such as task interference, data bias, or optimization saturation.
>
> **1.3 Fusion embedding layer**
>
> Additionally, we appreciate the reviewer’s attention to the fusion embedding layer. Our design introduces multimodal tokens into the original vocabulary to integrate discrete indices from the visual encoder (VQGAN). This includes 8,192 discrete tokens (from the VQ codebook) and two special tokens, `<START_IMG>` and `<END_IMG>`, forming the fusion embedding layer to seamlessly integrate visual information into the language model input. We will clarify and define this term in the method section.
>
>
> ## 2. Comparative Experiments
>
> We understand the reviewer’s request to compare our method with more powerful LVLMs for better evaluation. To address this, we add experiments with the enhanced **HealthGPT-XL** version and additional comparisons:
>
> Method|VQA-RAD|SLAKE|PathVQA|MMMU|OMVQA
> -|-|-|-|-|-
> Janus-Pro|62.9|51.3|48.9|54.1|32.7
> Emu3|72.1|57.3|54.1|28.7|39.5
> Qwen2-VL|73.3|61.9|62.6|46.0|59.5
> GPT-4o|57.4|54.3|66.5|48.7|36.7
> MedMax|74.9|86.8|91.6|27.3|95.1
> HealthGPT-M3|73.7|74.6|78.7|43.3|68.5
> HealthGPT-XL|79.1|85.7|92.4|58.0|77.2
>
> We carefully evaluate the concurrent work MedMax, which performs well, especially in OmniMedVQA. We have updated the experimental comparisons content in the manuscript.
>
> It is important to emphasize that HealthGPT aims to provide a unified, scalable framework adaptable to diverse tasks. It can seamlessly incorporate high-quality datasets like MedMax to boost downstream performance, and its design is **highly complementary to data-driven approaches** such as HuatuoGPT-Vision and MedMax.
>
> ## 3. Model Details
>
> **3.1 VQGAN**
>
> We appreciate the reviewer’s focus on the model's resource consumption and component compatibility, which are essential for the method's reproducibility and practicality. Below are our detailed responses:
>
> The VQGAN model, pretrained on OpenImages, is validate on three medical modalities: X-ray, CT, and MRI. The results demonstrate its strong performance in preserving key structural and lesion region information, with good fidelity and consistency:
>
> Modality|SSIM|PSNR|MSE
> -|-|-|-
> CT|93.95|36.80|14.17
> MRI|87.10|34.45|26.54
> X-Ray|86.80|33.97|29.41
>
> **3.2 Training details**
>
> Our method demonstrates strong resource efficiency and reproducibility. In the default configuration, the **M3 model** completes a full training cycle in about **35 hours** using **8 A100 GPUs**, which we consider reasonable given its performance across multiple tasks and the modular design.
>
> **3.3 Code Release Plan**
>
> We recognize the importance of open sourcing for the community and have nearly completed organizing the code, data processing scripts, model weights (M3, L14, and XL versions), and inference workflows. We will release the full codebase publicly to ensure reproducibility and ease of extension.
>
> Thank you again for your recognition and suggestions. We believe the response better highlights HealthGPT’s technical value and practical potential, and we hope it contributes to the development of medical multimodal models.

---

> > ### Comment · Reviewer_B8gX · 2025-04-01
> >
> > Thank you for your responses. Most of my concerns are addressed. I raised my score to 4.

---

> > > ### Author Response · Authors · 2025-04-02
> > >
> > > Thank you for raising the score. Your valuable suggestions greatly contribute to the quality of our manuscript. Thank you again for your precious time and valuable suggestions!

---

### Official Review · Reviewer_8ufe · 2025-03-09

**Overall Recommendation:** 3

**Summary:**

This paper introduces a medical vision and generation framework based on LVLM. This pipeline consists of autoregressive generation, hierarchical visual perception, and heterogenous knowledge adaption. The authors provide experiment results on different datasets and compare with other baselines.

**Claims And Evidence:**

Yes

**Essential References Not Discussed:**

No

**Experimental Designs Or Analyses:**

The experiment and analysis are fine.

**Methods And Evaluation Criteria:**

Yes

**Other Comments Or Suggestions:**

1, Figure 1 is too much and redundant, which is more distracted than being helpful.

2, Many commonly used formulation and equations do not need to re-phrased to be presented in the paper.

**Other Strengths And Weaknesses:**

Strengths:

1, This paper provides a comprehensive study from the method to the results.

2, The writing and presentation help readers to understand and follow.

3, They discuss the main challenges in the medical domain from Line 88 to Line 97.


Weakness:

1, I am concerned about the novelty, especially in the training process. They are using a very common and typical approach including feature alignment, LoRA-based plugin, and fine-tuning.

2, For the key part, which is the heterogenous knowledge adaption, the MOE+LoRA is also a well-used approach.

3, I understand that the application in medical domain with LLM might not be a traditional approach. With an extremely extensive experiments in a new domain plus a limited adjusted approach, I am concerned about the significance of contribution and if it is more like a benchmark work.

**Questions For Authors:**

1, Could you please provide some insights of the efficiency of the proposed approach?

2, Could you please emphasize which KEY part of your work is ORIGINALLY proposed by you? I appreciate if you could answer it in a simply and straightforward way. For example, "from Line xxx to Line xxx, I use xxx to solve the challenge of xxx".

**Relation To Broader Scientific Literature:**

They have the impact in the medical domain.

**Theoretical Claims:**

There are no proofs or theoretical claims in this paper.

---

> ### Author Rebuttal · Authors · 2025-03-30
>
> Thank you for your insightful review. Your insights have helped us clarify HealthGPT's contributions in unified architecture design, task modeling efficiency, and real-world application potential. Below are our point-by-point responses.
>
> # 1. Contribution Statement
>
> **1.1 Core Contribution**
>
> We appreciate the reviewer’s detailed focus on our method and contributions. It is worth noting that `Reviewers wUUr, ugJa, and B8gX` have recognized the novelty of our work, describing it as  `innovative` and `meaningful`. Building on this recognition, we would like to respectfully clarify a potential misunderstanding regarding the core contribution of our paper.
>
> - **Paradigm Innovation.**  Our primary innovation lies in **proposing the first unified Med-LVLM paradigm**, which goes beyond alignment or fine-tuning strategies. Specifically:
>   - At the **global-task level**, we enable the model to learn and handle heterogeneous characteristics and formats across both comprehension and generation.
>   - At the **local-task level**, we design mechanisms to efficiently learn and transfer shared knowledge across diverse sub-tasks.
> - **Method Innovation.** We carefully designe H-LoRA to align with our proposed paradigm, and further optimize TLS and HVP to support its integration. Thus, we emphasize that **alignment and fine-tuning are techniques we adopt to support this unified paradigm, not the main novelty itself**.
>
> To further clarify this point, we briefly summarize the key contributions:
> 1. Lines 139–142 (left): We propose the first unified Med-LVLM to solve the challenge of integrating diverse medical tasks under a single LVLM framework.
> 2. Lines 74–100(right): We introduce H-LoRA to solve the significant conflict between comprehension and generation tasks, greatly improving both the performance and efficiency over MoE+LoRA-based methods.
> 3. Lines 104(right)–126(left): We propose HVP and TLS to solve the challenge of heterogeneous knowledge fusion and hierarchical visual feature requirements.
> 4. Lines 127–136(left): we introduce VL-Health to solve the lack of high-quality datasets that jointly support unified tasks in the medical domain.
>
> Thus, this work goes beyond benchmarking by establishing a unified Med-LVLM framework that expands the capabilities of current medical models, which is crucial for advancing practical medical solutions.
>
> **1.2 H-LoRA batter than MoE+LoRA**
>
> Our results show that while MoE+LoRA improves performance, it violates the PEFT principle of efficiency. In contrast, H-LoRA significantly outperforms MoE+LoRA in both performance and training efficiency, as supported by theoretical analysis (`Reviewer wUUr (Section 2: H-LoRA)`) and experiments (Fig.5, Tab.5, Tab.10). H-LoRA achieves higher average scores (+3.6/+0.38 on comprehension/generation) with only 67% of the training time. We kindly hope these results clarify the necessity and advantages of H-LoRA.
>
> **1.3 Med-LVLM Significance**
>
> We appreciate the reviewer’s thoughtful concern regarding the application of Med-LLMs. As highlighted in prior studies [1,2], developing Med-LLMs is both a **timely scientific challenge and a transformative opportunity for healthcare**. These models can **bridge multimodal medical data, support informed decision-making, and ultimately improve clinical outcomes**.
>
> # 2. Efficiency Analysis
>
> We appreciate the reviewer’s focus on efficiency, vital in resource-limited medical scenarios. To validate efficiency, we compared H-LoRA with MoELoRA under identical settings. H-LoRA outperformed MoELoRA across tasks and reduced training time to 67% of MoELoRA’s:
>
> PEFT Method|LoRA|MoELoRA|HydraLoRA|H-LoRA
> -|-|-|-|-|
> TFLOP|311.6|204.9|238.3|313.7
>
> Besides, our three-stage learning strategy eliminates unnecessary padding caused by task sequence length mismatches, reducing training FLOPs by 22% (from 6.4e19 to 5.0e19) and lowering memory usage per GPU from 36GB to 30GB, saving 17%.
>
> Finally, HealthGPT-M3 completed training on 8 A100 GPUs in 35 hours, significantly fewer GPU hours than most general LVLMs, demonstrating its potential for rapid deployment in medical scenarios.
>
> # 3. Content Refinement
>
> We thank the reviewer for their helpful suggestions. We will simplify Figure 1 by streamlining labels, standardizing colors, and improving the layout to focus on structural logic.
>
> Regarding the formulas, we appreciate the reviewer pointing out redundancy. We have identified and will simplify or merge unnecessary functions and symbols (e.g., in Equations 4 and 5), ensuring the focus remains on our key innovations.
>
>
> We sincerely thank you for your valuable suggestions.  We believe this work addresses critical gaps in medical multimodal LVLMs and supports their deployment in real medical scenarios.
>
> **Reference**
>
> [1] A Survey of Large Language Models for Healthcare: from Data, Technology, and Applications to Accountability and Ethics
> [2] A Survey of Large Language Models in Medicine: Progress, Application, and Challenge

---

### Official Review · Reviewer_ugJa · 2025-03-12

**Overall Recommendation:** 4

**Summary:**

The paper introduces HealthGPT, a unified Medical Large Vision-Language Model (Med-LVLM) designed to integrate medical visual comprehension and generation. The proposed model employs innovative methods including Heterogeneous Low-Rank Adaptation (H-LoRA), hierarchical visual perception (HVP), and a three-stage training strategy (TLS). HealthGPT leverages a specially curated VL-Health dataset comprising multiple comprehension and generation tasks in medical imaging. Experimental results clearly demonstrate superior performance compared to state-of-the-art methods across diverse medical tasks such as modality conversion, super-resolution, and medical visual question answering (VQA).

**Claims And Evidence:**

The claims presented are robustly supported by experimental evidence, including thorough comparisons to state-of-the-art models across multiple benchmark tasks (Tables 1, 2, and 3). Performance metrics such as SSIM, PSNR, and MSE clearly substantiate the advantages of HealthGPT in both comprehension and generation scenarios.

**Essential References Not Discussed:**

While the paper adequately covers related works, including some additional emerging unified LVLM approaches or advanced MoE methods would further contextualize its contributions.

**Experimental Designs Or Analyses:**

The validity and soundness of the experimental designs are well-established, particularly for critical tasks such as modality conversion and super-resolution, which are backed by extensive benchmarks and standardized metrics.

**Methods And Evaluation Criteria:**

The proposed methods and evaluation criteria, including specific benchmarks such as VQA-RAD, SLAKE, PathVQA, and IXI datasets, are appropriate and well-suited to the addressed problems. The experimental methodology is carefully designed and rigorously validated.

**Other Comments Or Suggestions:**

The paper could improve clarity by explicitly discussing potential limitations and dataset biases.

**Other Strengths And Weaknesses:**

Strengths:

Originality in combining comprehension and generation capabilities in a single Med-LVLM.

Significant methodological innovations (H-LoRA, hierarchical visual perception).

Strong empirical validation against existing baselines.

Weaknesses:

Limited exploration of generalization capabilities across diverse medical scenarios not directly addressed in benchmarks.

**Questions For Authors:**

Can HealthGPT generalize effectively to medical conditions or modalities that were not explicitly included in the training datasets?

How well does HealthGPT handle complex multi-modal medical tasks involving simultaneous inputs from multiple modalities?

**Relation To Broader Scientific Literature:**

The paper clearly contextualizes its contributions within the broader literature, showing significant advancement over previous models like Med-Flamingo, LLaVA-Med, and Unified-IO 2.

**Theoretical Claims:**

The paper primarily addresses methodological advancements and does not involve explicit theoretical proofs.

---

> ### Author Rebuttal · Authors · 2025-03-30
>
> We greatly appreciate the reviewer’s recognition of our work. Your recognition of the model’s originality, innovation, and effectiveness greatly encourages us and has helped improve HealthGPT.  Below are our point-by-point responses.
>
> ## 1. Unified LVLM Comparison
>
> Thank you for the suggestion. Below, we provide further comparisons with SoTA unified LVLMs:
>
> Method|VQA-RAD|SLAKE|PathVQA|MMMU|OMVQA
> -|-|-|-|-|-
> chameleon|45.4|54.3|54.0|25.7|18.9
> Janus-Pro|62.9|51.3|48.9|54.1|32.7
> Emu3|72.1|57.3|54.1|28.7|39.5
> HealthGPT-M3|73.7|74.6|78.7|43.3|68.5
> HealthGPT-XL|79.1|85.7|92.4|58.0|77.2
>
> Meanwhile, we present an improved **HealthGPT-XL with enhanced performance**. We believe this additional experiment demonstrates significant advantages of our method.
>
> ## 2. Generalization Capabilities
>
> **2.1 Generalization ability**
>
> To demonstrate HealthGPT’s generalization and practical value, we highlight two aspects:
> 1. **Benchmark generalization**: The model performs well on unseen tasks and modalities (e.g., OmniMedVQA, MMMU-Med), showing robustness beyond the training distribution.
> 2. **Clinical collaboration**: Ongoing partnerships with public hospitals on ocular disease diagnosis have yielded promising early results, with further evaluation in progress.
>
> These results underscore HealthGPT’s potential for both benchmark-level generalization and practical clinical impact.
>
> **2.2 Complex medical tasks**
>
> Thank you for your thoughtful concerns. We recognize their importance and provide the following clarifications:
>
> 1. **Unseen medical conditions**: HealthGPT demonstrates strong generalization and medical knowledge coverage, enabling it to handle most previously unseen conditions.
> 2. **Unseen modalities**: In cases lacking prior modality-specific data, relying solely on a pre-trained LLM may introduce bias. However, our plugin-based learning approach allows for rapid adaptation using high-quality modality-specific data.
> 3. **Simultaneous multimodal inputs**: This engineering challenge can be effectively addressed through targeted data collection and training. We are actively collaborating with hospitals to tackle this issue in real-world clinical settings.
>
> We will incorporate the clarifications in the next version.
>
> ## 3. Differences with MoE Mechanism
>
> We appreciate the reviewer’s detailed attention to the model's structure, allowing us to clarify the **differences between H-LoRA and MoE architectures** and highlight the **necessity and advantages** of our design.
>
> Existing work explore LoRA and MoE combinations, including symmetric[1] and asymmetric structures[2]. While performance benefits are evident, we observe that introducing LoRA experts significantly increases resource consumption, compromising the efficiency of PEFT.
>
> To validate this, we compare the training efficiency of several LoRA+MoE structures with the same LoRA setting (r=64, k=4):
>
> Method|LoRA|MoELoRA|HydraLoRA|DS-LoRA|H-LoRA
> -|-|-|-|-|-
> TFLOP|311.6|204.9|238.3|224.1|313.7
>
> DS-LoRA is a LoRA mechanism based on the powerful MoE model Deepseek-V3. We find that MoELoRA, HydraLoRA, and DS-LoRA significantly reduce training speed, while H-LoRA avoids this limitation.
>
> Unlike traditional MoE scheduling, H-LoRA uses low-rank matrix subspace dynamic routing, preventing performance degradation and instability from issues like routing jitter and load imbalance. It also adaptively activates parameter subspaces, creating soft isolation and weak coupling to enhance adaptability to task heterogeneity.
>
> Additionally, we emphasize that our approach is not a dense-to-MoE alternative for expanding FFN. While traditional MoE increases capacity by expanding parameters, our method focuses on **balancing performance and computational resources** in efficient fine-tuning.
>
> ## 4. Other Issues
>
> **4.1 Potential Limitations**
>
> We appreciate the reviewer’s attention to potential limitations. Our work explores the feasibility of efficiently handling unified comprehension and generation tasks in medical scenarios. As shown in Figure 2, joint training faces a significant bottleneck in data-scarce medical tasks. However, task cooperation enhancement is a long-term focus, and we aim to explore the complementary potential between the two tasks for synergistic effects.
>
> **4.2 Dataset Bias**
>
> We understand that bias in sample distribution and task labeling can impact model generalization. To improve chat-LVLM practicality, we balanced general instruction data with curated medical datasets during training, enhancing generalization and medical knowledge. A section on dataset bias has been added to the appendix.
>
> Thank you again for your positive and encouraging feedback and insightful comments. We hope this response further clarifies our design, contributions, and the model's potential.
>
> **Reference**
>
> [1] When MOE Meets LLMs: Parameter Efficient Fine-tuning for Multi-task Medical Applications
>
> [2] HydraLoRA: An Asymmetric LoRA Architecture for Efficient Fine-Tuning

---

### Official Review · Reviewer_wUUr · 2025-03-12

**Overall Recommendation:** 3

**Summary:**

This paper presents HealthGPT, a medical large vision-language model (Med-LVLM) that unifies both comprehension and generation of medical images. The key contributions include:
1. Heterogeneous Low-Rank Adaptation (H-LoRA);
2. Hierarchical Visual Perception ;
3. Three-Stage Learning Strategy;
4. A curated multi-modal dataset covering 7 comprehension tasks (VQA, medical reasoning, pathology) and 5 generation tasks (modality transformation, super-resolution, image reconstruction);
5. The model outperforms existing Med-LVLMs (e.g., LLaVA-Med, Med-Flamingo, HuatuoGPT-Vision) and achieves state-of-the-art (SOTA) results across multiple medical imaging benchmarks.

While the work is technically solid and introduces meaningful innovations, its improvements in VQA performance are marginal, and the evaluation of image generation lacks perceptual metrics.

**Claims And Evidence:**

1. HealthGPT is the first unified Med-LVLM integrating comprehension and generation: Supported, as existing works (e.g., LLaVA-Med) focus on comprehension only.
2. H-LoRA effectively separates comprehension and generation tasks: Partially supported. Ablation studies demonstrate benefits, but there is no clear comparison with other PEFT methods (e.g., MoELoRA) beyond training efficiency.
3. HealthGPT achieves SOTA performance on medical VQA and generation tasks: Not supported. The improvements in VQA over HuatuoGPT-Vision are minor, and some of the published work has a higher VQA effect than this study.
4. Three-stage training improves performance: Well-supported, as the structured training prevents catastrophic forgetting.

Suggested Improvements:
1. Provide qualitative visualizations for VQA (e.g., heatmaps, model reasoning breakdown).
2. Add perceptual metrics (e.g., FID, LPIPS) for evaluating generation tasks.
3. For QA/VQA, please compare with more advanced methods such as LLAVA-MED+ +, etc.

**Essential References Not Discussed:**

None.

**Experimental Designs Or Analyses:**

The experiments are well-structured but have notable limitations:
1. VQA evaluation lacks detailed error analysis and no comparison with the SOTA methods, like  LLAVA-MED+ +.
2. No human evaluation for image generation quality.

Suggested Improvements:
1. Include failure case analysis for VQA errors.
2. Conduct blind radiologist assessments of generated medical images.
3. Please compare with more advanced methods such as LLAVA-MED+ +, etc.

**Methods And Evaluation Criteria:**

The methodology is well-designed:
1. H-LoRA: Decouples learning across tasks using a Mixture-of-Experts (MoE)-like mechanism.
2. HVP: Dynamically selects abstract vs. detailed image features.
3. TLS: Ensures a structured alignment between vision-language modalities.

However, the evaluation metrics for generation tasks are too simplistic (SSIM, PSNR, MSE) and VQA performance shows only limited improvement over previous works.

Suggested Improvements:
1. Introduce qualitative evaluations (e.g., radiologist evaluations of generated images).
2. Add error analysis for VQA failures.

And the other question is that although this approach takes into account both VQA and image generation, which is also a feature of this study, it seems that the two tasks cannot improve the quality of each other.

**Other Comments Or Suggestions:**

No.

**Other Strengths And Weaknesses:**

Strengths:
1. Strong multi-modal integration of comprehension and generation.
2. Innovative H-LoRA adaptation mechanism.
3. New VL-Health dataset, covering diverse medical tasks.

Weaknesses:
1. Limited VQA improvement over prior Med-LVLMs, and not compared with LLaVA-Med++.
2. Image generation evaluation lacks perceptual metrics (e.g., FID, LPIPS).
3. Comparison to MoELoRA only focuses on training time, not task performance.
4. Although this method takes into account both text and image output, it seems that the two tasks cannot improve the quality of each other.

**Questions For Authors:**

1. Has HealthGPT been tested in real-world clinical settings?
2. Could you provide qualitative examples where VQA fails?

**Relation To Broader Scientific Literature:**

The paper builds on prior Med-LVLMs (LLaVA-Med, Med-Flamingo) but fails to compare against general vision-language models (e.g., SEED, Chameleon) and the other models like LLaVA-Med++.

**Theoretical Claims:**

While the paper does not present new theoretical derivations, H-LoRA’s mathematical formulation could be further elaborated:
1. There is no direct theoretical comparison to standard LoRA and MoELoRA.
2. The routing mechanism in H-LoRA is not well justified.

Suggested Improvement: Provide a theoretical explanation for why H-LoRA is expected to generalize better than standard LoRA in medical contexts.

---

> ### Author Rebuttal · Authors · 2025-03-30
>
> We sincerely thank the reviewer for the detailed and thoughtful feedback, which greatly helped us refine our experiments and methodology, and better highlight HealthGPT’s innovation and practical value.  Below are our point-by-point responses.
> ## 1. VQA Analysis
> **1.1 Comparison with LLaVA-Med++**
>
> We wish to clarify that **LLaVA-Med++ is not a chat LVLM and is fine-tuned on individual downstream tasks**, which compromises its generalization ability. For fairness, we use LLaVA-Med++'s training setting and the results are below:
>
> Method|VQA-RAD|SLAKE|PathVQA
> -|-|-|-
> LLaVA-Med++|86.0/77.1|85.3/80.8|98.9/58.7
> HealthGPT-M3|85.6/71.9|85.9/81.7|99.1/67.2
>
> **1.2 Comparison experiments**
>
> Notably, we further release the **HealthGPT-XL** with a stronger LLM (Qwen2.5-32b). **We believe this model is the new SOTA Med-LVLM compared with baselines**:
>
> Method|VQA-RAD|SLAKE|PathVQA|MMMU|OMVQA
> -|-|-|-|-|-
> chameleon|45.4|54.3|54.0|25.7|18.9
> Janus-Pro|62.9|51.3|48.9|54.1|32.7
> RadFM|58.3|34.4|58.4|31.3|36.2
> LLaVA-Med++ (SFT)|64.7|87.1|55.1|x|x
> Med-MoE|66.9|52.6|69.1|32.7|55.8
> HealthGPT-M3|73.7|74.6|78.7|43.3|68.5
> HealthGPT-XL|79.1|85.7|92.4|58.0|77.2
>
> The aforementioned results further demonstrate HealthGPT's effectiveness.
>
> **1.3 Failure case**
>
> After analysis, the failure cases stem from two sources:
> 1. Broad questions (e.g., 'What is present?')
> 2. Same question, different answers (Q → A1 vs. Q → A2)
>
> Addressing (1) and (2) typically requires fine-tuning on the corresponding training set. However, our model still achieves the best performance through strong generalization and instruction-following capabilities.
>
> **1.4 VQA qualitative**
>
> We visualize heatmaps for different questions and masked key regions to assess answer quality: https://anonymous.4open.science/r/HealthGPT-9533/visualization.png. Results confirm our model relies on critical regions for reasoning.
>
> ## 2. H-LoRA
> **2.1 Performance of H-LoRA**
>
> We believe the reviewer may have misunderstood—we did compare performance with MoELoRA. Fig.5, Tab.4, and Tab.10 in our paper systematically demonstrate the **performance and efficiency advantages of H-LoRA**.
>
> **2.2 H-LoRA better than LoRA**
>
> - **Efficiency.** Existing research confirms that LoRA with MoE and router mechanisms improves performance for medical tasks [1]. To enhance efficiency, we propose internalizing LoRA experts as matrix space separation: $\sum ka w_i A_i B_i / r \rightarrow ka A \odot \mathcal{W} B / r$.
>
> - **Effectiveness.** We find the conventional router mechanism reduces the expected scaling factor from $a/r$ to $a/(rk)$, leading to performance degradation [2]. We correct the scaling factor and experimentally validate its effectiveness in our paper. Additionally, we included an ablation study on the router module:
>
> Method|VQA-RAD|SLAKE|PathVQA|MMMU|OMVQA
> -|-|-|-|-|-
> HealthGPT-M3 w/o router|72.2|70.7|76.2|38.0|66.4
> HealthGPT-M3|73.7|74.6|78.7|43.3|68.5
>
> We hope the above analyses and experiments further demonstrate the potential of H-LoRA in the medical domain.
>
> ## 3. Image Metrics
> **3.1 LPIPS & FID**
>
> We appreciate the reviewer’s concern about perceptual consistency in images. As requested, we’ve added LPIPS and FID metrics (https://anonymous.4open.science/r/HealthGPT-9533/lpips_fid.png), showing HealthGPT’s strength in both pixel-level and perceptual quality.
>
> **3.2 Human evaluation**
>
> We also conduct human evaluation of the modality conversion task across five dimensions, comparing against the best-performing BBDM method. H-LoRA achieves **higher average scores** (4.30/4.52 vs. BBDM's 3.54/4.16):https://anonymous.4open.science/r/HealthGPT-9533/human_eval.png.
>
> ## 4. Other Issues
> **4.1 Mutual Gains between Comprehension and Generation**
>
> We would like to respectfully clarify that our work aims to **mitigate the significant conflicts between comprehension and generation tasks**, rather than assuming inherent complementarity. Given current modeling paradigms and data limitations in the medical domain, achieving mutual benefit is particularly challenging (see Fig. 2). To this end, we design dedicated mechanisms to reduce such conflicts and enable unified modeling.
>
> **4.2 Clinical Potential**
>
> We highly value clinical evaluation and are already collaborating with two public hospitals, which provide ocular disease reports and case data for validation. These efforts represent a concrete step toward assessing the clinical applicability of our approach, and preliminary evaluations on the provided clinical data show **promising advantages of our model in ocular disease understanding**.
>
> **4.3 Dataset Bias**
>
> Dataset bias is an important concern. Due to space limits, please see our response to `Reviewer ugJa (Section 4: Dataset Bias)`.
>
> We hope our response clarifies the model’s rationale and its potential in medical applications.
>
> **Reference**
>
> [1] When MOE Meets LLMs: Parameter Efficient Fine-tuning for Multi-task Medical Applications
> [2] A Rank Stabilization Scaling Factor for Fine-Tuning with LoRA

---

> > ### Comment · Reviewer_wUUr · 2025-04-08
> >
> > OK, thank you for the author's response. In my view, this result is mainly driven by the capabilities of qwen2.5. Another point is that the model addresses the text-to-image and image-to-text problems, but it does not test interleaved data, which is significant. Based on the feedback from all reviewers, I will maintain this score for now.

---

> > > ### Author Response · Authors · 2025-04-08
> > >
> > > We sincerely thank the reviewer for carefully feedback.
> > >
> > > **(1) Regarding the Source of Performance Improvement**
> > >
> > > We greatly appreciate the reviewer’s attention to the foundation model issue. We clarify that **our three models of different scales (M3/L14/XL) all achieve SoTA performance across multiple medical benchmarks**, demonstrating that our framework effectively adapts to and extends models of varying scales. Notably, even our smallest 3.8B model outperforms LLaVA-Med++ in zero-shot evaluations and achieves leading results among current SoTA medical LVLMs. Therefore, **the use of the Qwen2.5 foundation aims to pursue better performance, while it is our systematic innovation that fundamentally drives the achievement of SoTA results**.
> > >
> > > We have previously experimented with the same foundation model as LLaVA-Med++ (LLaMA3-8B) and **achieve superior performance**. However, as this version performs similarly to HealthGPT-M3 with twice the parameters, we do not adopt it. The experimental results are as follows:
> > >
> > > Method (zero-shot)|VQA-RAD|SLAKE|PathVQA|MMMU|OMVQA
> > > -|-|-|-|-|-
> > > LLaVA-Med++|64.7|87.1|55.1|x|x
> > > HealthGPT-LLaMA3-8B|74.1|80.2|73.5|41.3|65.7
> > >
> > > We further clarify the motivation for training HealthGPT-XL with Qwen2.5. In the medical domain, where tasks require deeper knowledge and higher reasoning precision, adopting a stronger foundation model is a natural and necessary response to domain-specific demands. Benefiting from the **plug-in design of the HealthGPT framework**, we flexibly adapt to foundations of different scales and capabilities without compromising pre-trained medical knowledge. The successful application of HealthGPT-XL not only meets the foundational requirements of medical tasks but also **validates the scalability and compatibility of our method**, demonstrating HealthGPT’s potential for continuous evolution and broad applicability across various pre-trained foundations.
> > >
> > > We hope these clarifications fully address the reviewer’s concerns, highlight the innovation and application value of our work, and emphasize that **the performance gains stem from the proposed method rather than merely relying on a stronger foundation model**.
> > >
> > > **(2) Regarding the evaluation of interleaved text-image inputs**
> > >
> > > We thank the reviewer for highlighting the important issue of evaluating interleaved text-image inputs.
> > >
> > > First, we would like to clarify that most existing medical LVLMs have not yet adapted to interleaved input capabilities, resulting in a lack of fair baselines. Moreover, no standardized benchmark currently exists in the medical domain for such evaluations, making it difficult to accurately assess potential performance improvements through interleaved modeling in the short term.
> > >
> > > Nevertheless, we highly value the reviewer’s suggestion and **are actively extending our framework to support interleaved inputs**:
> > > - Comprehension: we construct a multi-image dataset by inserting image tokens into multi-turn dialogues, enabling simple, efficient, and compatible interleaved inputs within our training paradigm.
> > > - Generation: we organize multiple tasks into an interleaved format while ensuring compatibility and rigor in the reasoning process.
> > >
> > > We sincerely appreciate the reviewer’s constructive feedback. We clarify that the limited demand for frequent interleaving in medical tasks stems from the domain’s high standards of rigor, stability, and accuracy, not from limitations of our method. Interleaving is common in general multimodal tasks designed for large-scale instruction-following, whereas medical applications have fundamentally different requirements.
> > >
> > > We sincerely hope the reviewer **understands the unique modeling needs of the medical domain and does not lower the evaluation of our contributions due to differences in interaction formats or application objectives.** Meanwhile, we actively respond to the feedback, continue improving our interleaved modeling capabilities, and promptly update our experimental progress.
> > >
> > > **(3) Final Remarks**
> > >
> > > We fully respect the reviewer’s decision to maintain the current score at this stage.
> > >
> > > At the same time, we emphasize that unified comprehension and generation for medical tasks—still highly challenging and underexplored compared to general domains—requires significant exploration and experimentation. Our work **proposes the first framework specifically tailored for medical applications, focusing on preserving pre-trained medical knowledge and enhancing LVLMs’ comprehension and generation through a plug-in design**. Despite challenges such as limited data and stringent application standards, we strive to deliver a **simple, scalable, and effective solution**.
> > >
> > > **We sincerely hope that, based on our methodological innovations and the critical value of our work in the medical domain, the reviewer will recognize our contributions and consider a more positive evaluation.**
> > >
> > > Once again, we sincerely thank the reviewer for the opportunity to improve our work.

---

### Decision · Program_Chairs · 2025-05-01

**Decision:**

Accept (spotlight poster)

**Comment:**

This submission proposes HealthGPT, a unified medical large vision-language model (Med-LVLM), integrating comprehension and generation tasks in medical imaging through a novel heterogeneous low-rank adaptation (H-LoRA), hierarchical visual perception (HVP), and a three-stage learning strategy. It contributes the comprehensive VL-Health dataset covering diverse comprehension and generation tasks. Overall, the reviewers found the methodological innovations significant, particularly the H-LoRA approach, and recognized the strong empirical validation against existing baselines. Initial concerns about incremental improvements in VQA performance, the lack of perceptual evaluation metrics for generation, and limited interpretability were effectively addressed by the authors in their detailed rebuttal and supplementary analyses. Reviewers noted the work’s substantial practical potential and the thoroughness of its experimental evaluation. While one reviewer raised concerns about novelty and whether the contribution aligns fully with ICML's typical methodological focus, the extensive experiments and the practical significance for medical applications justified acceptance. Thus, the committee decided to accept this paper, highlighting its robust innovation, strong empirical validation, and considerable potential for practical medical applications.